# Global profiling of regulatory elements in the histone benzoylation pathway

Duo Wang[1,2,5], Fuxiang Yan[1,2,5], Ping Wu[3,5], Kexue Ge[1,2], Muchun Li[1,2,4], Tingting Li[1,2], Ying Gao[1], Chao Peng [3✉] & Yong Chen [1,2,4✉]

Lysine benzoylation (Kbz) is a recently discovered post-translational modification associated with active transcription. However, the proteins for maintaining and interpreting Kbz and the physiological roles of Kbz remain elusive. Here, we systematically characterize writer, eraser, and reader proteins of histone Kbz in *S. cerevisiae* using proteomic, biochemical, and structural approaches. Our study identifies 27 Kbz sites on yeast histones that can be regulated by cellular metabolic states. The Spt-Ada-Gcn5 acetyltransferase (SAGA) complex and NAD$^+$-dependent histone deacetylase Hst2 could function as the writer and eraser of histone Kbz, respectively. Crystal structures of Hst2 complexes reveal the molecular basis for Kbz recognition and catalysis by Hst2. In addition, we demonstrate that a subset of YEATS domains and bromodomains serve as Kbz readers, and structural analyses reveal how YEATS and bromodomains recognize Kbz marks. Moreover, the proteome-wide screening of Kbz-modified proteins identifies 207 Kbz sites on 149 non-histone proteins enriched in ribosome biogenesis, glycolysis/gluconeogenesis, and rRNA processing pathways. Our studies identify regulatory elements for the Kbz pathway and provide a framework for dissecting the biological functions of lysine benzoylation.

[1] State Key Laboratory of Molecular Biology, Shanghai Institute of Biochemistry and Cell Biology, Center for Excellence in Molecular Cell Science, Chinese Academy of Sciences, 200031 Shanghai, China. [2] University of Chinese Academy of Sciences, 100049 Beijing, China. [3] National Facility for Protein Science in Shanghai, Zhangjiang Lab, Shanghai Advanced Research Institute, Chinese Academy of Science, 201210 Shanghai, China. [4] School of Life Science and Technology, ShanghaiTech University, 201210 Shanghai, China. [5]These authors contributed equally: Duo Wang, Fuxiang Yan, Ping Wu. ✉email: pengchao@sari.ac.cn; yongchen@sibcb.ac.cn

In eukaryotes, histone post-translation modifications (PTMs) play crucial roles in gene transcription, DNA repair, DNA replication, cell metabolism, and chromatin structural organization[1–4]. Lysine benzoylation (Kbz) is a newly discovered histone modification initially identified in mammalian cells[5]. Histone benzoylation mainly occurs on the N-terminal tails of histones and has a different genome distribution than Kac[5]. The levels of histone benzoylation are significantly stimulated upon sodium benzoate treatment[5,6]. Because sodium benzoate has been widely used as an antimicrobial preservative in the food and cosmetics industry[7,8], the identification of histone benzoylation sheds light on how a preservative may alter the human epigenome and affect the gene expression profile[5]. However, the underlying regulatory mechanism of histone benzoylation and the associated physiological consequence remain poorly defined, partially due to the lack of knowledge of the proteins responsible for depositing, removing, and reading Kbz marks.

Histone acylation can be spontaneously deposited by chemically reactive metabolites or catalyzed by enzymes (e.g., histone acetyltransferases). Whether histone benzoylation is driven by a non-enzymatic or an enzymatic mechanism is not known. Which enzymes are involved in the deposition of Kbz is not yet investigated. Moreover, the initial screening of histone deacetylases (HDACs) identified SIRT2 to be the only HDAC that has histone debenzoylase activity[5], but a recent study suggested that SIRT1 could also serve as a debenzoylase[9]. How SIRT1 or SIRT2 recognizes and removes benzoylation marks remains unclear.

A recent study showed that a subset of DPF (double plant homeodomain finger) and YEATS (Yaf9, ENL, AF9, Taf14, and Sas5) domains, but not bromodomains, serve as readers for histone Kbz in vitro, and structural analyses revealed molecular mechanisms for Kbz readout by DPF and YEATS domains[6]. However, whether DPF- or YEATS-mediated Kbz recognition is conserved across species and whether any other domain may be involved in Kbz recognition have not been thoroughly investigated.

Mounting evidence shows that lysine acylation could occur on non-histone proteins[10,11]. Acetylome studies have revealed a tremendous number of acetylation sites in thousands of proteins in human and yeast cells[12–15]. Dynamic changes in acetylation have been closely linked to protein stability, protein interaction, enzyme activity, and protein localization[14,15]. Other forms of lysine acylation, including succinylation (Ksucc), 2-hydroxyisobutyrylation (Khib), β-hydroxybutyrylation (Kbhb), and malonylation (Kmal), are also commonly identified on non-histone proteins[10,11,16,17]. These acylation-modified proteins were associated with metabolic processes, DNA repair, spliceosome, ribosome, and RNA processing[3,10,11,16,17]. To date, the protein substrates bearing Kbz and their functions in cellular biological processes remain unknown, thereby hindering the understanding of the broad biological functions of lysine benzoylation.

Here, we systematically characterize the regulatory proteins of histone benzoylation in S. cerevisiae and reveal the molecular mechanism by which these proteins write, erase and read Kbz marks through genetic, biochemical, and structural analyses. Beyond conserved histone benzoylation, whole-cell proteomic profiling of Kbz modification expands the protein substrates containing Kbz to a wide range of proteins, especially proteins involved in ribosome biogenesis and metabolic processes. These results lay the foundation to dissect the roles of lysine benzoylation in diverse cellular processes.

## Results

### Identification of histone Kbz sites in yeast.
To confirm that Kbz is an evolutionarily conserved histone mark, we first examined whether Kbz modification was present in S. cerevisiae by immunoblotting using a pan anti-Kbz antibody[5]. The specificity of this antibody was first evaluated by a dot blot assay (Supplementary Fig. 1a), in which the pan anti-Kbz antibody can recognize only peptides bearing a benzoyl lysine but not peptides containing lysine acetylation (Kac) or lysine crotonylation (Kcr) marks. The Kbz signal was readily detected on core histones extracted from logarithmically growing yeast cells but not on recombinant Xenopus laevis histone octamer purified from E.coli and cell lysates from E.coli (Fig. 1a).

Consistent with the previous reports that sodium benzoate stimulates histone benzoylation in mammalian cells[5,7], we found that sodium benzoate treatment significantly enhanced the Kbz and Kac signals on yeast histones in a dose-dependent manner (Fig. 1b). We then used liquid chromatography with tandem mass spectrometry (LC-MS/MS) to measure the cellular concentration of benzoyl-CoA in yeast cells treated with 10 mM sodium benzoate for 6 h. The determined concentration of benzoyl-CoA was approximately 0.33 μM (296 ng/ml) (Supplementary Fig. 1b), which is in the similar concentration ranges of crotonyl-CoA, malonyl-CoA, and propionyl-CoA in yeast and mammalian cells[18–20]. Thus benzoate-stimulated benzoyl-CoA could serve as an amble source for lysine benzoylation.

Previous studies have demonstrated that the levels of histone acylation are sensitive to metabolic states[3,21]. To investigate whether cellular metabolism also regulates the histone Kbz level, we compared the histone Kbz levels in the presence of different carbon sources. The traditional yeast culture medium with 2% glucose yielded the strongest Kbz signals (Fig. 1c). Substitution of glucose with raffinose, galactose, or ethanol substantially decreased the Kbz signals on histones (Fig. 1c). This suggests that histone benzoylation is a modification responsive to the carbon source, reinforcing the link between histone acylation and carbohydrate metabolism[3,21].

To map the Kbz sites on yeast histones, we performed proteomic analysis using histones extracted from yeast cells treated with 10 mM sodium benzoate for 6 h. Although Kbz was a low-abundance modification and only 0.056% of histone peptides were benzoylated, we identified 27 Kbz sites on yeast H3, H4, H2A, H2A.Z, and H2B (Fig. 1d). Notably, nine Kbz sites were detected in samples without sodium benzoate treatment (Fig. 1d). The representative MS/MS spectra of four peptides (H3K9bz, H3K14bz, H3K27bz, and H4K44bz) are shown in Fig. 1e–h, and others are shown in Supplementary Fig. 2. Most of the yeast histone Kbz sites are localized on the flexible N-terminal tails of core histones, while there are 5 Kbz sites located in the globular domains (H2BK88, H3K42, H3K56, H3K115, and H4K44), differing from the distribution pattern of Kbz sites in mammalian cells that are exclusively located on the N-terminal tails of core histones[5]. This indicates that the Kbz modification in yeast may have different functional roles from that in mammalian cells. Collectively, these results suggest that histone benzoylation is an evolutionarily conserved mark and that the benzoylation level in yeast is sensitive to carbohydrate metabolism.

### The Gcn5-containing complex is a histone benzoyltransferase.
To gain insights into the dynamic regulation of Kbz, we sought to identify the writers, erasers and readers for Kbz. We first searched for potential histone benzoyltransferases through a candidate screening approach. Histone acetyltransferases (HATs) can deposit diverse acyl groups onto histone lysines and are potential candidates for histone benzoyltransferases. We examined the histone Kbz levels in yeast strains in which seven non-essential HATs were deleted individually. Western blot analyses by a pan anti-Kbz antibody showed that only Gcn5 deletion substantially

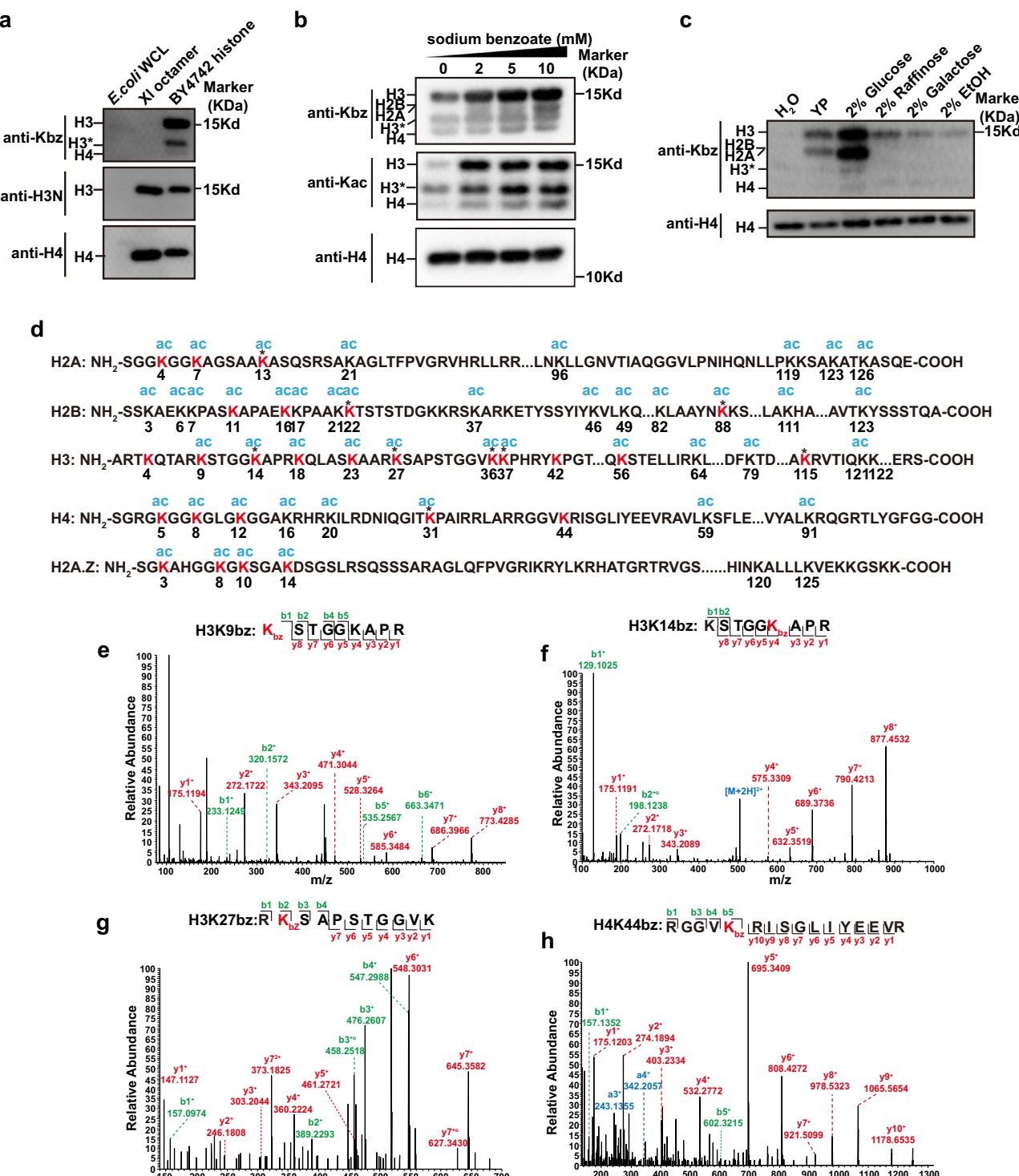

**Fig. 1 Identification of histone Kbz in yeast. a** Detection of the Kbz signals from core histones extracted from BY4742 yeast cells. The specificity of pan anti-Kbz was confirmed because it did not recognize any band from the *E.coli* whole-cell lysate (*E.coli* WCL) or recombinant *Xenopus laevis* histone octamer (Xl octamer) purified from *E.coli*. The H3* represents clipping H3. Source data for panels **a–c** are provided in the Source Data file. **b** Sodium benzoate treatment significantly enhanced the Kbz and Kac levels on yeast histones in a dose-dependent manner. BY4742 cells were grown in YPD to log phase, and then sodium benzoate was added into the medium for 6 h, followed by extraction of histones and western blotting analysis. **c** The level of histone Kbz was regulated by the type of carbohydrate. BY4742 cells were grown in standard medium with 2% glucose to log phase and then transferred to H₂O or other indicated mediums for 4 h, followed by extraction of histones and western blotting analysis. **d** Illustrations of all Kbz sites identified on core histones extracted from yeast cells treated with 10 mM sodium benzoate. The detected Kbz sites are shown in red. Nine Kbz sites detected in samples without sodium benzoate treatment are labeled by asterisks. Most Kbz sites overlap with the known Kac sites, except for H3K42, H3K115, and H4K44. **e–h**: MS/MS spectra of H3 and H4 peptides bearing Kbz modification. (**e**) H3K9; (**f**) H3K14; (**g**) H3K27; (**h**) H4K44. Predicted b- and y-type ions are listed above and below the peptide sequence, respectively. The circle symbol indicates the neutral loss of water. Matched ions are labeled in the spectra.

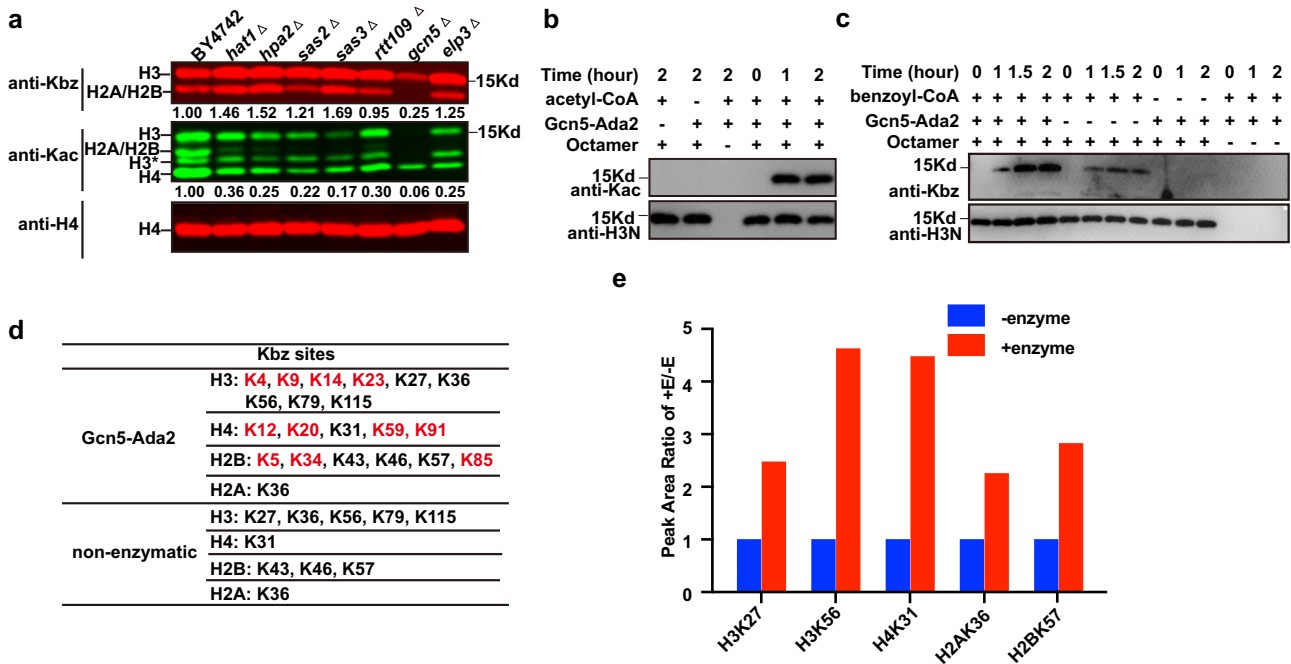

**Fig. 2 Gcn5-containing complex is a histone benzoyltransferase. a** Fluorescence quantitative western blot analysis comparing the Kbz and Kac levels in WT (BY4742) and HAT-deletion strains. Total H4 was used as the loading control. The fluorescence signals from mutant strains were normalized by setting the WT fluorescence signal to 1.0 and were labeled in the image. Source data for panels (**a**–**c**) are provided in the Source Data file. **b** The Ada2-Gcn5 subcomplex could catalyze acetylation of the histone octamer in vitro. The reaction products were detected by western blot analysis using a pan-Kac antibody. **c** Histone benzoylation can occur by enzymatic and non-enzymatic mechanisms. The reaction products in different combinations were detected by western blot analysis using a pan-Kbz antibody. **d** The list of LC-MS/MS-identified histone benzoylation sites deposited by the non-enzymatic mechanism and by Gcn5-Ada2. The unique sites catalyzed by Gcn5-Ada2 are labeled red. **e** Gcn5-Ada2 catalyzed histone benzoylation more efficiently than spontaneous reaction. The **Y**-axis represents the ratio of MS/MS peak areas under enzymatic and non-enzymatic conditions. This result is from a single experiment.

decreased the overall histone Kbz signals compared with the WT strain, implying that Gcn5 may be responsible for catalyzing lysine benzoylation (Fig. 2a). Except for Gcn5, deletion of any other HATs caused an unexpectedly mild increase in the Kbz signal (Fig. 2a). Whether this is due to experimental variation and what is the underlying mechanism for such a Kbz increase need further investigation. In addition to non-essential HATs, we also tested whether Esa1p, an essential HAT, could catalyze histone benzoylation. Since the esa1Δ strain is not viable, we utilized the temperature-sensitive mutant esa1-531. The acetylation levels of all histones were completely abolished in the esa1-531 mutant at the non-permissive temperature (37 °C). In sharp contrast, the Kbz level of H3 was only slightly reduced, and the Kbz levels of H2A/H2B were severely decreased at the non-permissive temperature, indicating that Esa1 might have weak benzoyltransferase activity specific to H2A/H2B (Supplementary Fig. 3a).

To demonstrate whether Gcn5 can directly catalyze histone Kbz, we performed an in vitro assay using the recombinant Gcn5-Ada2 subcomplex purified from E.coli and the endogenously-purified SAGA complex (Supplementary Fig. 3b). The Gcn5-Ada2 binary complex is the minimum HAT module in the SAGA complex[22,23]. In the in vitro assay system, both the recombinant Ada2-Gcn5 subcomplex and the endogenous SAGA complex could catalyze the acetylation and benzoylation of histone octamers (Fig. 2b, c and Supplementary Fig. 3c, d). Moreover, the SAGA complex had higher catalytic activity than the Ada2-Gcn5 subcomplex in both acetylation and benzoylation reactions, suggesting that additional SAGA components fine-tune the acylation activity. It should be noted that the benzoylation activity of Gcn5-Ada2 is much weaker than the acetylation activity (Supplementary Fig. 3e, f), preventing us from measuring

the kinetic parameters of the Gcn5-catalyzed benzoylation reaction.

To further provide structural insights into Gcn5-mediated histone benzoylation, we modeled a benzoyl-CoA-bound Gcn5-Ada2 structure based on the structures of the succinyl-CoA-bound hGCN5 complex (PDB 5TRL)[24] and yeast Gcn5-Ada2 complex (PDB 6CW2)[25] (Supplementary Fig. 3g). Benzoyl-CoA was built in a conformation seen in the succinyl-CoA-hGcn5 complex, except for replacing the terminal succinyl group with a benzoyl group. In the modeled structure (Supplementary Fig. 3h), the pyrophosphate group and 3′-phosphate-adenosine moiety of CoA sit in a positively charged pocket, while the benzoyl group and pantetheine moiety are docked to a relatively hydrophobic groove of Gcn5, supporting the hypothesis that the active site of the Gcn5 complex has adequate space to accommodate the large benzoyl group (Supplementary Fig. 3h, i). Consistent with the structural model, the Gcn5-Ada2 complex has comparable benzoyl-CoA and succinyl-CoA binding affinities (Supplementary Fig. 3j). Together both in vivo and in vitro evidence demonstrate that Gcn5 could function as a histone benzoyltransferase in yeast.

In addition to enzyme-catalyzed reactions, lysine acylations can also be non-enzymatically deposited, spontaneously occurring between highly electrophilic acyl-CoAs and nucleophilic lysine residues. To examine whether histone benzoylation could be driven by a non-enzymatic mechanism, we incubated the recombinant Xenopus laevis histone octamer purified from E.coli with benzoyl-CoA in the absence of enzymes. Without enzymes, lysine benzoylation signals also appeared, although the signals were much weaker than those in the presence of Gcn5-Ada2 (Fig. 2c). In sharp contrast, spontaneous acetylation was

negligible under our assay conditions (Fig. 2b). We then used LC-MS/MS to map the benzoylation sites deposited by non-enzymatic mechanism or catalyzed by Gcn5-Ada2(Supplementary Fig. 3k). LC-MS/MS analyses confirmed that histone benzoylation could spontaneously occur on histones in the absence of enzymes. Meanwhile, Gcn5-Ada2 could induce more benzoylated histone sites and catalyze the benzoylation at a higher abundance than that without enzymes on specific sites (Fig. 2d, e and Supplementary Data 1). Collectively, these data reveal that histone benzoylation could occur by both enzyme-dependent and enzyme-independent mechanisms.

**A subset of HDACs are histone debenzoylases**. Next, we aimed to search for the enzyme(s) that could remove the benzoyl group from histones. We hypothesized that certain histone deacetylases (HDACs) would be potential debenzoylases. There are three groups of HDACs in yeast: Class I (Rpd3), Class II (Hda1, Hos1, Hos2, Hos3), and Class III sirtuin family (Sir2, Hst1, Hst2, Hst3, Hst4). We first compared the levels of histone benzoylation and acetylation between BY4742(WT) and ten HDAC-deletion mutants by fluorescence quantitative western blot analyses of core histones extracted from yeast cells (Fig. 3a). Most HDAC-deletion strains except $sir2\Delta$ significantly increased the global histone acetylation signals more than 2-fold, underpinning the importance of these enzymes in erasing acetylation marks (Fig. 3a). In sharp contrast, deletion of most HDACs only mildly increased the benzoylation level less than 2-fold (Fig. 3a). Removal of Hst2 caused the highest increase in the Kbz signal by approximately 1.8-fold (Fig. 3a). Unexpectedly, deletions of some HDACs, including Rpd3, Hos3, and Sir2, decreased the global histone benzoylation levels (Fig. 3a), and the underlying mechanism needs further investigation[5].

To confirm the debenzoylation activity catalyzed by Hst2, whose deletion led to the highest increase in Kbz signals, we purified the recombinant Hst2 protein from *E.coli* and performed an in vitro assay using a synthetic H3K9bz peptide (ARTKQTAR-Kbz-STGGKAPRKQY) as the substrate. The molecular weight change from the debenzoylation reaction was monitored by matrix-assisted laser desorption ionization-time of flight (MALDI-TOF) mass spectrometry. Figure 3b clearly shows that Hst2 can catalyze debenzoylation of the H3K9bz peptide with a molecular weight decrease of 104 Da, although the catalytic rate of debenzoylation is significantly slower than that of deacetylation (Fig. 3b, c). To quantitatively compare the activities of debenzoylase and deacetylase, we performed kinetic experiments using H3K9ac and H3K9bz peptides, respectively. The results showed that the $K_m$ for deacetylation was approximately 6-fold higher than that for debenzoylation, while the turnover number ($k_{cat}$) for deacetylation was 63.5-fold higher than that for debenzoylation. Therefore, the catalytic efficiency ($k_{cat}/K_m$) for deacetylation was approximately 10.6 times higher than that for debenzoylation (Fig. 3d, e). As a negative control, the purified Hos3 protein could efficiently remove the acetyl group but not the benzoyl group from modified peptides (Supplementary Fig. 4a, b). Collectively, these results support the conclusion that Hst2 could act as a histone debenzoylase.

Next, we sought to profile the substrate specificity of Hst2-mediated debenzoylation. We synthesized a panel of histone peptides containing different benzoylation sites on histone H3. The progression of the debenzoylation reaction was monitored by time-resolved MALDI-TOF mass spectrometry. Hst2 could remove the benzoylation marks from all the peptides tested, albeit with different efficiencies (Fig. 3f). We also performed isothermal titration calorimetry (ITC) to quantify the interactions between Hst2 and different Kbz peptides. The ITC results demonstrated that all the peptides could bind to Hst2 with

dissociation constants ($K_d$) ranging from 1.4 to 15 μM (Fig. 3g, h). The binding affinity between Hst2 and each substrate was highly correlated with the corresponding debenzoylation activity. The strongest binding was observed between Hst2 and H3K9bz ($K_d = 1.39$ μM), consistent with the highest debenzoylase activity on H3K9bz (Fig. 3f, g). H3K14bz and H3K23bz had the lowest binding affinities with Hst2; correspondingly, Hst2 had the lowest debenzoylase activities on H3K14bz and H3K23bz. Subsequent sequence alignment indicated a critical arginine residue adjacent to the target lysine site determining the tight binding between Hst2 and histone substrates (H3K9, H3K18, and H3K27) (Fig. 3h). The presence of positively charged residues downstream of the target lysine also seems to be required for strong binding with Hst2. We speculate that Hst2 prefers the target lysine flanked by positivity-charged residues (R/K). Taken together, our in vivo and in vitro experiments suggest that Hst2 serves as one of the erasers of lysine benzoylation with a broad substrate spectrum.

**The structural basis of H3K9bz recognition by Hst2**. To elucidate the molecular mechanism of Hst2-mediated debenzoylation, we determined the crystal structure of Hst2$_{8-294}$ in the complex with the benzoylated H3 peptide (H3$_{5-14}$K9bz) at 2.0 Å resolution (Fig. 4a, Supplementary Fig. 4c, and Table 1). We also tried to crystallize the ternary complex composed of Hst2, benzoylated substrate, and cofactor NAD$^+$, but the solved complex structure at 1.8 Å only showed the density of 2'-O-benzoyl-ADP-ribose, the cofactor product (Fig. 4b, Supplementary Fig. 4d, e). We presumed that the ternary complex initially formed and reacted, producing 2'-O-benzoyl-ADP-ribose that remained bound while the debenzoylated substrate and nicotinamide dissociated. The appearance of 2'-O-benzoyl-ADP-ribose also suggests that the debenzoylation process utilizes a similar catalytic mechanism as deacetylation catalyzed by NAD$^+$-dependent HDACs (Supplementary Fig. 4f).

In both structures, Hst2$_{8-294}$ exhibits an elongated structure containing a large Rossmann-fold domain and a small zinc-containing domain as described previously (Fig. 4a, b)[26]. The large Rossmann domain is composed of ten α-helices (α1, α2, α6-8, α10-14) and six parallel β-strands (β1-3, β7-9). The small domain consists of four α-helices (α3-5, α9) and three β-strands (β4-6). Both the H3 peptide and the cofactor product bind to the cleft between two domains (Fig. 4b). The superimposition of two structures by the large domains shows a 17° rotation of the small domain and the rearrangement of the α2–α3 loop involved in contacting the 2'-O-benzoyl-ADP-ribose (Fig. 4b). Such rigid body rotation might be facilitated by four glycine residues serving as hinges located at junctions between the large domain and the small domain (Fig. 4a).

In the Hst2-peptide structure, histone H3 residues from T6 to K14 were modeled based on the electron density (Supplementary Fig. 4c). The histone peptide adopts a bent conformation and makes extensive contacts with Hst2 through both electrostatic and hydrophobic interaction (Fig. 4c, d). The majority of H3 binds to a negatively charged surface of Hst2 (Fig. 4c), explaining the preference of benzoylated lysine flanked by positively charged residues (Fig. 3h). Specifically, H3 R8 is positioned in an acidic patch containing Hst2 D68 and D70 (Fig. 4c), reinforcing the notion that the strong histone binders of Hst2 (H3K9, H3K18, and H3K27) share the RK signature (Fig. 3h). Consistent with the structural model, the H3 R8A mutation decreased the binding affinity between Hst2 and H3K9bz and weakened the debenzoylation activity of Hst2 (Fig. 4e, f). Accordingly, the Hst2 D68A/D70A mutation reduced the interaction with H3K9bz and impaired the debenzoylase activity of Hst2, confirming the importance of

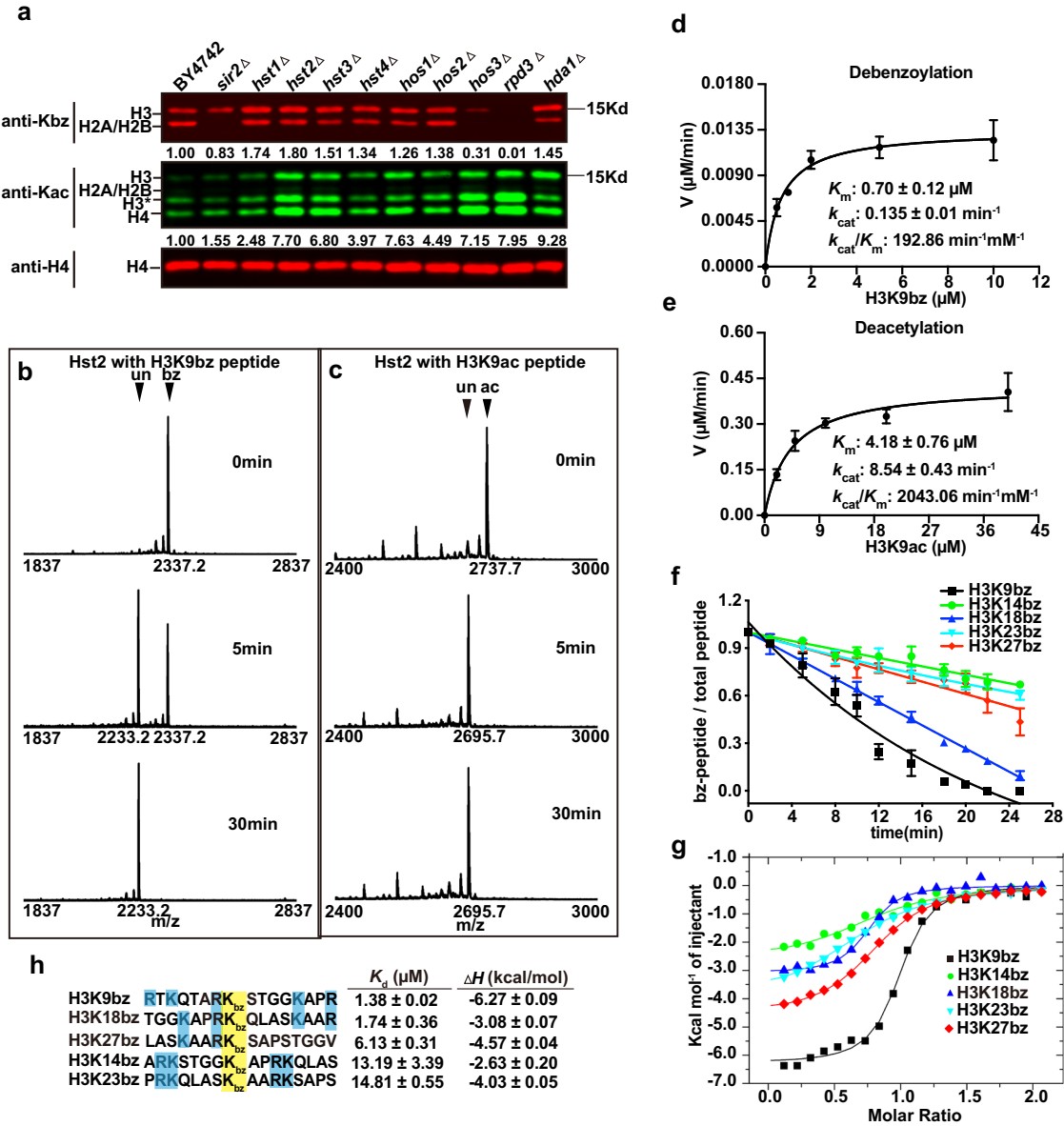

**Fig. 3 Hst2 is a histone debenzoylase. a** Fluorescence quantitative western blot analysis comparing the Kbz and Kac levels in WT (BY4742) and ten HDAC-deletion strains. The fluorescence signals from mutant strains were normalized by setting the WT fluorescence signal to 1.0 and were labeled in the image. The H3* represents clipping H3. Source data are provided in the Source Data file. **b** Debenzoylase activity of Hst2$_{FL}$ protein (2 μM) with H3K9bz peptide (10 μM) shown by the representative MALDI-TOF spectra at 0, 5, and 30 min. The peaks for H3K9bz (bz, $m/z$ 2,337) and unmodified (un, $m/z$ 2,233) products are labeled. **c** Deacetylase activity of Hst2$_{FL}$ protein (0.5 μM) with H3K9ac peptide (10 μM) shown by the representative MALDI-TOF spectra at 0, 5, and 30 min. The peaks for H3K9ac (ac, $m/z$ 2,737) and unmodified (un, $m/z$ 2695) products are labeled. **d** The Michaelis-Menten plot for Hst2 fixed at 0.1 μM with varied H3K9bz peptide concentrations. Data are presented as mean ± SD, $n = 3$ biological independent measurements. Source data for panels (**d**, **e**) are provided in the Source Data file. **e** The Michaelis–Menten plots for Hst2 fixed at 0.05 μM with varied H3K9ac peptide concentrations. Data are presented as mean ± SD, $n = 3$ biological independent measurements. **f** Hst2-mediated debenzoylation reaction monitored by time-resolved MALDI-TOF mass spectrometry. The Y-axis represents the percentage of Kbz peptide in total peptides calculated from the MALDI-TOF spectra. Data are presented as mean ± SD, $n = 3$ biological independent measurements. Source data are provided in the Source Data file. **g** ITC binding curves for Hst2 interaction with different Kbz peptides. **h** Sequence alignment of benzoylated H3 peptides used in the current study. The alignment is centered on the target lysine (yellow). Positively charged R/K residues are shown in blue. The dissociation constants ($K_d$) and enthalpy changes (ΔH) derived from the ITC assay (panel **g**) are shown.

electrostatic interactions in coordinating the recognition of histone substrates with Hst2 (Fig. 4e, f). Interestingly, a series of Hst2 residues (D68, D70, D187, D190, E194, E236, E237) form an acidic ring on the surface of Hst2 for basic histone binding, and these acidic residues are not conserved in other class III HDACs (Supplementary Fig. 4k), suggesting that other class III HDACs may have different substrate specificities from Hst2.

Benzoylated lysine 9 (K9bz) is snugly inserted into a relatively hydrophobic tunnel (Fig. 4d) and adopts the same geometric configuration of H4K16ac as seen in the Hst2-H4K16ac structure (Fig. 4g). In H3K9bz- and H4K16ac-bound structures, both aliphatic side chains of lysine residues form the same van der Waals interactions with Hst2 H135, V182, F184, L188, and V228. The backbone amides of H3K9 and H4K16 identically coordinate

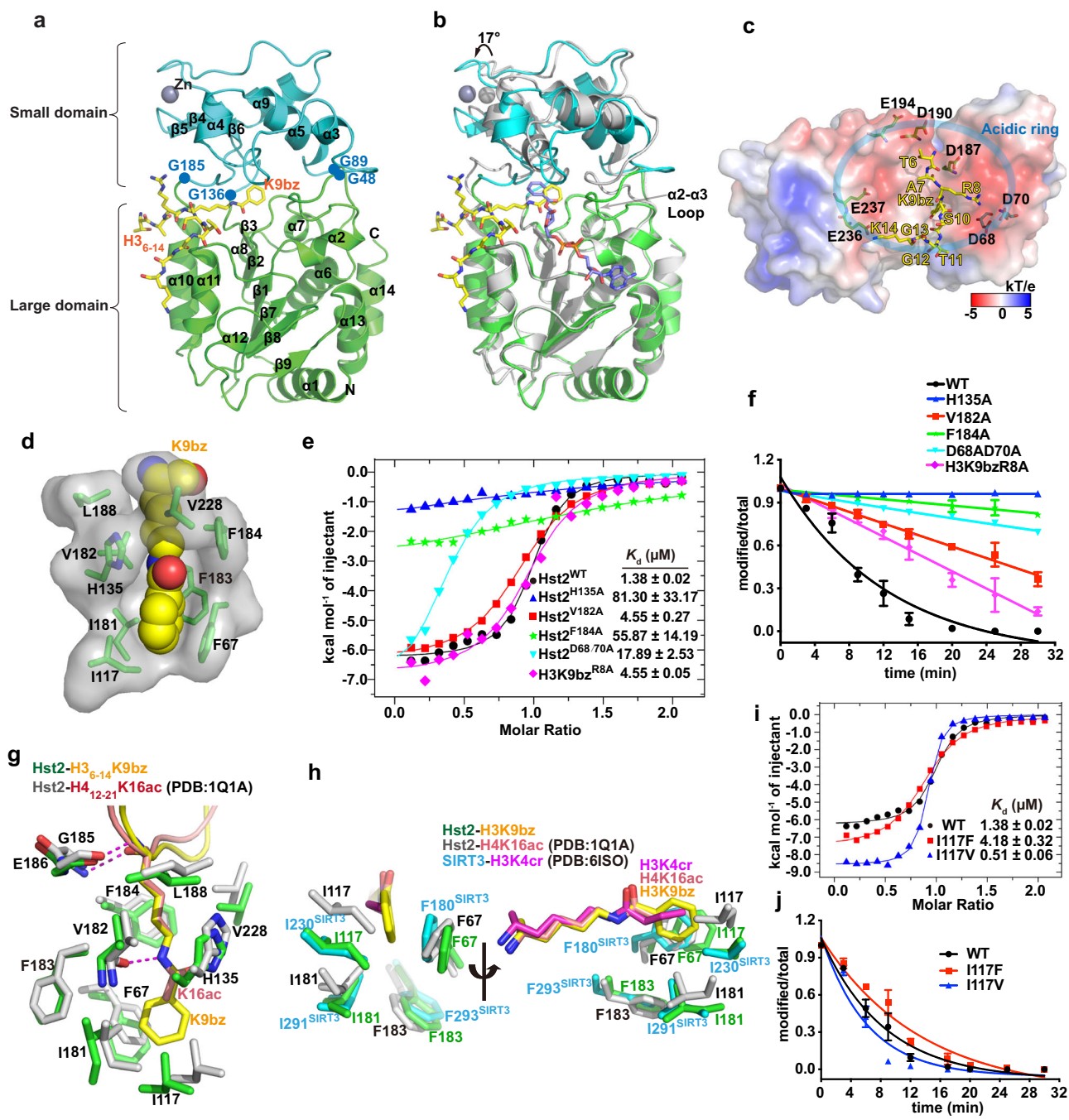

two hydrogen bonds to the backbones of G185 and E186, and the amides of two acyl groups form a hydrogen bond to V182 (Fig. 4g). The almost identical Hst2-Kbz and Hst2-Kac interfaces contribute to the similar binding affinities of H3K9bz ($K_d = 1.38\,\mu M$) and H3K9ac ($K_d = 1.33\,\mu M$) to Hst2 (Fig. 4e and Supplementary Fig. 4g). Mutations of these hydrophobic residues to alanine, including H135, V182, and F184, reduced the interactions with both the H3K9bz-containing peptide and the H3K9ac-containing peptide to varying degrees: H135A and F184A severely disrupted the interaction with acylated peptides, while V182A mildly decreased the binding by approximately 3-fold. Consequently, these disruptive mutations abrogated the debenzoylation and deacetylation activities of Hst2 (Fig. 4e, f and Supplementary Fig. 4g, h).

We next examined the structural difference between Kbz and Kac binding with Hst2, focusing on the binding pockets for the

methyl group in the acetyl moiety and the benzene ring in the benzoyl moiety. The methyl group and benzene ring are surrounded by the same set of hydrophobic residues F67, I117, I181, and F183 (Fig. 4h). The bulky benzene ring of K9bz induces a 20° rotation of the phenyl group of F67 to coordinate a π-π stacking interaction with the benzene ring (Fig. 4h). Additionally, K9bz induces shifts of I117 and I181 to generate a larger space to accommodate the benzene ring (Fig. 4d). A similar configuration of the crotonyl-binding pocket of SIRT3 composed of F180, F293, I230, and I291 was also observed in the SIRT3-H3K4cr structure (PDB: 6ISO)[27] (Fig. 4h). We reason that the subtle configuration rearrangement of these residues (F67, I117, I181) ensures the various acylation mark recognition by Hst2.

To further support the importance of these residues in clamping the benzene ring, we explored the possibility of engineering Hst2 by mutating I117 to create an Hst2 mutant

**Fig. 4 The structural basis for H3K9bz recognition by Hst2. a** The overall structure of the Hst2$_{8-294}$–H3$_{6-14}$K9bz complex. The large domain and small domain in Hst2 are shown in green and cyan, respectively. H3K9bz peptide is presented as a stick-model (yellow). Four glycine residues located at the junctions between the large and small domains are labeled. **b** Superimposition of Hst2-H3K9bz and Hst2-2'-O-benzoyl-ADP-ribose complexes indicates the relative 17° rotation of the small domain and the rearrangement of the α2-α3 loop. The large domain and small domain in the Hst2-H3K9bz complex are shown in green and cyan, respectively. Hst2 in the Hst2-2'-O-benzoyl-ADP-ribose complex is shown in gray. The cofactor product 2'-O-benzoyl-ADP-ribose is shown in slate. **c** The interface between Hst2 and H3 peptide. Hst2 is shown as a surface model colored according to its electrostatic potential (positive potential: blue; negative potential, red). H3 residues from T6 to K14 are shown in yellow stick models. An acidic ring formed by a series of acidic residues is labeled. **d** The H3K9bz binding pocket of Hst2. The side chain of K9bz (shown as a ball model) is inserted into a relatively hydrophobic tunnel of Hst2. **e** ITC measurements reveal that Hst2 mutations and H3K9bz$^{R8A}$ weaken the Hst2–H3K9bz interaction. The dissociation constant ($K_d$) and their fitting errors are shown. **f** MALDI-TOF-based debenzoylase assays show that Hst2 mutations and H3K9bz$^{R8A}$ decrease the debenzoylase activity of Hst2. Data are presented as mean ± SD, $n = 3$ biological independent measurements. Source data are provided in the Source Data file. **g** Comparison of the almost identical Kbz and Kac-binding pockets in Hst2-H3K9bz and Hst2-H4K16ac (PDB: 1Q1A) structures. H3K9bz is shown in yellow, and H4K16ac is shown in red. Hydrogen bonds are shown as magenta dashed lines. **h** Comparison of the acyl-group binding pockets by superimposition of Hst2-H3K9bz, Hst2-H4K16ac, and SIRT3-H3K4cr structures. The benzoyl, acetyl, and crotonyl groups are surrounded by four conserved hydrophobic residues. **i** ITC results reveal that Hst2 I117F and I117V mutants decrease and increase binding with H3K9bz peptides, respectively. The dissociation constant ($K_d$) and their fitting errors are shown. **j** MALDI-TOF-based debenzoylase assays show that Hst2 I117F and I117V mutants decrease and increase debenzoylase activities on H3K9bz peptides, respectively. Data are presented as mean ± SD, $n = 3$ biological independent measurements. Source data are provided in the Source Data file.

| Table 1 Data collection and refinement statistics for crystal structures. | | | | | |
|---|---|---|---|---|---|
| | Hst2$_{8-294}$-cofactor | Hst2$_{8-294}$-peptide | Taf14$_{2-137}$-H3K9bz | Sas5$_{2-139}$-H3K27bz | Sth1$_{1248-1359}$-H3K14bz |
| *Data collection* | | | | | |
| Space group | P2$_1$2$_1$2$_1$ | P2$_1$2$_1$2$_1$ | P6$_5$ | P3$_2$21 | P1 |
| *Cell dimensions* | | | | | |
| a, b, c (Å) | 42.17, 67.36, 92.51 | 42.15, 68.40, 94.51 | 113.79,113.79,26.62 | 126.58,126.58,48.71 | 23.62, 33.21, 42.45 |
| α, β, γ (°) | 90.00, 90.00, 90.00 | 90.00, 90.00, 90.00 | 90.00,90.00,120.00 | 90.00,90.00,120.00 | 88.14, 83.72, 70.48 |
| Wavelength (Å) | 0.9785 | 0.9785 | 0.9785 | 0.9785 | 0.9785 |
| Resolution (Å) | 50-2.0 | 50-1.8 | 50-2.0 | 50-2.4 | 50-1.4 |
| $R_{merge}$ | 0.074(0.347)$^a$ | 0.063(0.802) | 0.094(0.733)$^a$ | 0.08(0.899) | 0.053(0.321)$^a$ |
| I/σI | 23.4(3.7) | 26.8(1.4) | 21.2(2.0) | 25.0(2.0) | 40.4(4.5) |
| Completeness (%) | 99.9(99.3) | 98.9(89.9) | 98.7(87.5) | 99.9(100) | 92.5(90.2) |
| Redundancy | 6.4(5.7) | 6.3(4.5) | 9.2(7.2) | 6.7(6.9) | 7.0(6.2) |
| *Refinement* | | | | | |
| Resolution (Å) | 28.0-2.0 | 28.6-1.8 | 24.1-2.0 | 27.4-2.4 | 25.2-1.4 |
| No. reflections | 18404 | 25193 | 12859 | 17665 | 22032 |
| $R_{work}$/$R_{free}$ (%) | 19.0/22.0 | 18.4/21.6 | 17.6/20.4 | 22.7/27.4 | 16.4/18.2 |
| *No. atoms* | | | | | |
| Protein | 2238 | 2276 | 1148 | 2329 | 1042 |
| Ligand | 45 | 18 | 17 | 45 | 23 |
| Water | 158 | 234 | 90 | 85 | 128 |
| *B-factors (Å$^2$)* | | | | | |
| Protein | 37.3 | 31.6 | 34.51 | 67.34 | 26.57 |
| Ligand | 27.9 | 34.1 | 44.91 | 59.55 | 33.34 |
| Water | 41.5 | 36.5 | 41.72 | 52.25 | 36.16 |
| *R.m.s. deviations* | | | | | |
| Bond lengths (Å) | 0.004 | 0.006 | 0.011 | 0.004 | 0.006 |
| Bond angles (°) | 0.761 | 1.593 | 1.042 | 0.741 | 0.910 |
| *Ramachandran plot statistics* | | | | | |
| Favored (%) | 96.85 | 97.48 | 99.26 | 97.78 | 100.00 |
| Allowed (%) | 3.15 | 2.52 | 0.74 | 2.22 | 0.00 |
| Outliers (%) | 0.00 | 0.00 | 0.00 | 0.00 | 0.00 |

$^a$Highest resolution shell is shown in parentheses.

showing altered specificity to benzoylated peptides. We identified an I117F mutant that decreased the binding affinity with K9bz by approximately 3~fold but slightly enhanced the binding with K9ac (Fig. 4i and Supplementary Fig. 4i); accordingly, Hst2 I117F mutant had decreased debenzoylase activity and slightly increased deacetylase activity (Fig. 4j and Supplementary Fig. 4j). This finding is consistent with the bulky phenylalanine residue partially occluding the fitting of the benzene ring due to steric clash but increasing the hydrophobic contact with the methyl group in the acetyl moiety. In addition, another mutant, I117V, which may expand the hydrophobic pocket to better accommodate the large benzene ring, led to a 2.6-fold binding enhancement with K9bz but not to an affinity change with K9ac (Fig. 4I, j and Supplementary 4i, j). Thus, our structural and biochemical studies demonstrate that Hst2 can function as a debenzoylase and identify residues essential for benzoyl recognition and removal, providing a powerful tool to further dissect the functions of Hst2 as a debenzoylase in cells.

**YEATS domains from Sas5 and Taf14 have weak Kbz binding activities**. The functional interpretation of histone acylation depends on chromatin-associating proteins harboring conserved reader domains, including bromodomain, YEATS, DPF, and the double pleckstrin homology (PH) domain[28]. A recent study revealed that the YEATS and DPF domains, but not the bromodomain, are readers for histone benzoylation in human cells[6]. The budding yeast has three YEATS-containing proteins but no DPF domain identified. To screen the potential benzoylation readers in yeast, we purified recombinant YEATS domains of Taf14, Sas5, and Yaf9 from *E. coli*. We then performed ITC binding assays to quantify the binding affinity between YEATS domains and five Kbz-containing H3 peptides (H3K9, K14, K18, K23, and K27). ITC analyses showed that both Taf14 and Sas5 have Kbz binding abilities, albeit with different specificities. Sas5$_{YEATS}$ prefers binding with H3K14 and H3K27, whereas Taf14 showed comparatively strong binding toward H3K9bz, K18bz, and K27bz (Fig. 5a, b). In sharp contrast, Yaf9$_{YEATS}$ exhibited no detectable binding to any tested Kbz peptide under our assay conditions (Fig. 5c), although Yaf9$_{YEATS}$ could bind to H3K27ac peptide with a similar $K_d$ ($K_d = 280\,\mu M$) as reported previously[29] (Supplementary Fig. 5a). These results demonstrate that the yeast YEATS domains from Taf14 and Sas5 are potential readers for benzoylated lysines and show different substrate preferences in vitro.

It should be noted that, although the interactions between Taf14/Sas5$_{YEATS}$ and Kbz peptides are relatively weak with $K_d$ in the range of several hundred micromolar, these binding affinities are comparable to those of yeast YEATS domains binding acetylated or crotonylated lysine (e.g., Taf14$_{YEATS}$ to K9ac, $K_d = 334\,\mu M$; Taf14$_{YEATS}$ to K9cr, $K_d = 75\,\mu M$)[30]. We also confirmed that Taf14$_{YEATS}$ had similar binding affinities to H3K9bz and H3K9cr by ITC assays (Supplementary Fig. 5b). Considering the established functional roles of Taf14$_{YEATS}$ binding H3K9 acetylation and crotonylation[31,32], our results thus attest to the physiological relevance of weak H3K9bz recognition by Taf14$_{YEATS}$. Moreover, Sas5$_{YEATS}$ showed a binding preference for Kbz over Kac (Supplementary Fig. 5c), suggesting that Sas5$_{YEATS}$ might serve as a Kbz reader in yeast cells.

**Structural basis for Kbz recognition by YEATS domains from Sas5 and Taf14**. To elucidate the molecular basis of Kbz recognition by the YEATS domains from Taf14$_{YEATS}$ and Sas5$_{YEATS}$, we determined the crystal structures of the Taf14$_{YEATS}$-H3$_{5-13}$K9bz and Sas5$_{YEATS}$-H3$_{18-37}$K27bz complexes (Table 1 and Supplementary Fig. 5d–g). In the 2.0 Å Taf14$_{YEATS}$-H3$_{5-13}$K9bz structure, H3 residues from Q5 to K9bz could be unambiguously built (Supplementary Fig. 5d, e). The overall configuration and binding mode of Taf14$_{YEATS}$ with H3K9bz are almost identical to those of H3K9ac-, H3K9cr-, and H3K9bu-bound Taf14$_{YEATS}$ complexes (Supplementary Fig. 5h, i)[32–34]. In the 2.4 Å Sas5-H3K27bz structure, all H3 residues from K18 to K36 have well-defined electron density, including an extra tyrosine residue (Y37) at the C-terminus designed for concentration determination by UV spectrometry (Supplementary Fig. 5f, g). The H3K27bz peptide binds to Sas5 in a partially similar configuration as the H3K27bz peptide in the hYEATS2-H3K27bz complex. Two H3K27bz peptides overlap at the target lysine (K27) and the flanking residues (A25-A31), but diverge at both the N- and C-termini (Supplementary Fig. 5j).

Taf14$_{YEATS}$ and Sas5$_{YEATS}$ adopt an immunoglobin-like β-sandwich fold (Fig. 5d). Both histone peptides are docked on top of the β-sandwich and are surrounded by three loops L$_{1-2}$, L$_{4-5}$, and L$_{6-7}$ (Fig. 5d). Superimposition of Taf14$_{YEATS}$-H3K9bz and Sas5$_{YEATS}$-H3K27bz revealed no obvious conformational

difference between the two YEATS domains, except for the L$_{1-2}$ loop region that makes contacts with the extended C-terminus of the H3 peptide in the Sas5$_{YEATS}$-H3K27bz complex (Fig. 5d). In the Taf14$_{YEATS}$-H3K9bz structure, H3 R8 forms salt bridges with Taf14 D104 (Supplementary Fig. 5k). Mutation of R8 severely disrupted the interaction between Taf14$_{YEATS}$ and H3K9bz (Supplementary Fig. 5b), reinforcing the importance of the neighboring arginine in ensuring binding specificity between Taf14$_{YEATS}$ and histone peptides containing RK signatures (H3K9, H3K18, and H3K27) (Fig. 5a). In the Sas5$_{YEATS}$-H3K27bz structure, in addition to K27bz-mediated interactions, H3 A29 and P30 make extensive hydrophobic contacts with a hydrophobic groove composed of Sas5 L28, R31, W82, and F108 (Supplementary Fig. 5l). The downstream AP signatures are only observed in H3K14 and H3K27 peptides, explaining the relatively higher binding affinities of Sas5$_{YEATS}$ with H3K14 and H3K27 peptides than other peptides (Fig. 5b).

Although H3K9bz and H3K27bz peptides show opposite orientations and their lysine side chains adopt different kinked configurations, their benzoyl groups are inserted into a conserved binding pocket sharing nearly identical interacting networks (Fig. 5e). The lysine side chains and the benzoyl groups are embraced by two arms: one arm is composed of a pentapeptide (residues 59–63 in Taf14 and residues 60-64 in Sas5), and the other arm is composed of a GWG motif that is the most conserved motif in all YEATS domains (Supplementary Fig. 5m). The carbonyl oxygen of the benzoyl group coordinates two hydrogen bonds with Taf14$^{W81}$/Sas5$^{W82}$ and Taf14$^{G82}$/Sas5$^{G83}$. The amine nitrogen of the benzoyl group forms one hydrogen bond with the conserved Taf14$^{T61}$/Sas5$^{S62}$, which also coordinates a hydrogen bond to an invariant histidine (Taf14$^{H59}$/Sas5$^{H60}$). A π-π-π stacking interaction composed of Taf14$^{W81}$/Sas5$^{W82}$, the benzene ring of Kbz, and Taf14$^{F62}$/Sas5$^{F63}$ dominantly determines the specific recognition of benzoylated lysine by YEATS domains (Fig. 5e). Mutations of aromatic residues involved in π-π-π stacking (Sas5 W82A and Taf14 W81A) completely disrupted the interaction with benzoylated peptides (Fig. 5f, g).

One recent study reported the crystal structures of mammalian AF9-H3K9bz and YEATS2-H3K27bz complexes[6]. Structural alignments of four complex structures reveal a universal molecular mechanism of Kbz recognition by YEATS domains across species (Fig. 5h)[6]. The key residues recognizing benzoylated lysine and interaction networks are well conserved, including the aromatic sandwich cage and the hydrogen bonds mediated by the benzoyl amide groups (Fig. 5h). However, the residues at the tip regions of the benzoyl-binding pocket, including Taf14$^{V29}$/Sas5$^{L30}$/AF9$^{F28}$/YEATS2$^{S230}$ and Taf14$^{A63}$/Sas5$^{K64}$/AF9$^{P63}$/YEATS2$^{P264}$, are different in the four complex structures. Superimposition of Taf14 structures in complex with H3K9ac, H3K9cr, H3K9bu, and H3K9bz also shows that although the tip residues V29 and A63 do not directly contact acylated lysine, they participate in defining the overall shape and size of the acylation-binding pocket, therefore affecting the binding preference with different types of acylation marks (Fig. 5i). We noticed that Yaf9 has a histidine residue at the corresponding position of Taf14$^{V29}$/Sas5$^{L30}$, which might partially explain the low affinity between Yaf9$_{YEATS}$ and Kbz peptides. The relatively bulky and positively charged environment created by histidine might not be conducive to binding the benzoyl group, which was confirmed by the impaired binding between the Sas5 L30H mutant and K27bz peptide (Fig. 5f). Thus, it is conceivable that these tip residues, which are not conserved in YEATS domains (Supplementary Fig. 5m), fine-tune the binding affinities and specificities with different acylation marks. Further mutagenesis studies confirmed this hypothesis. The Sas5

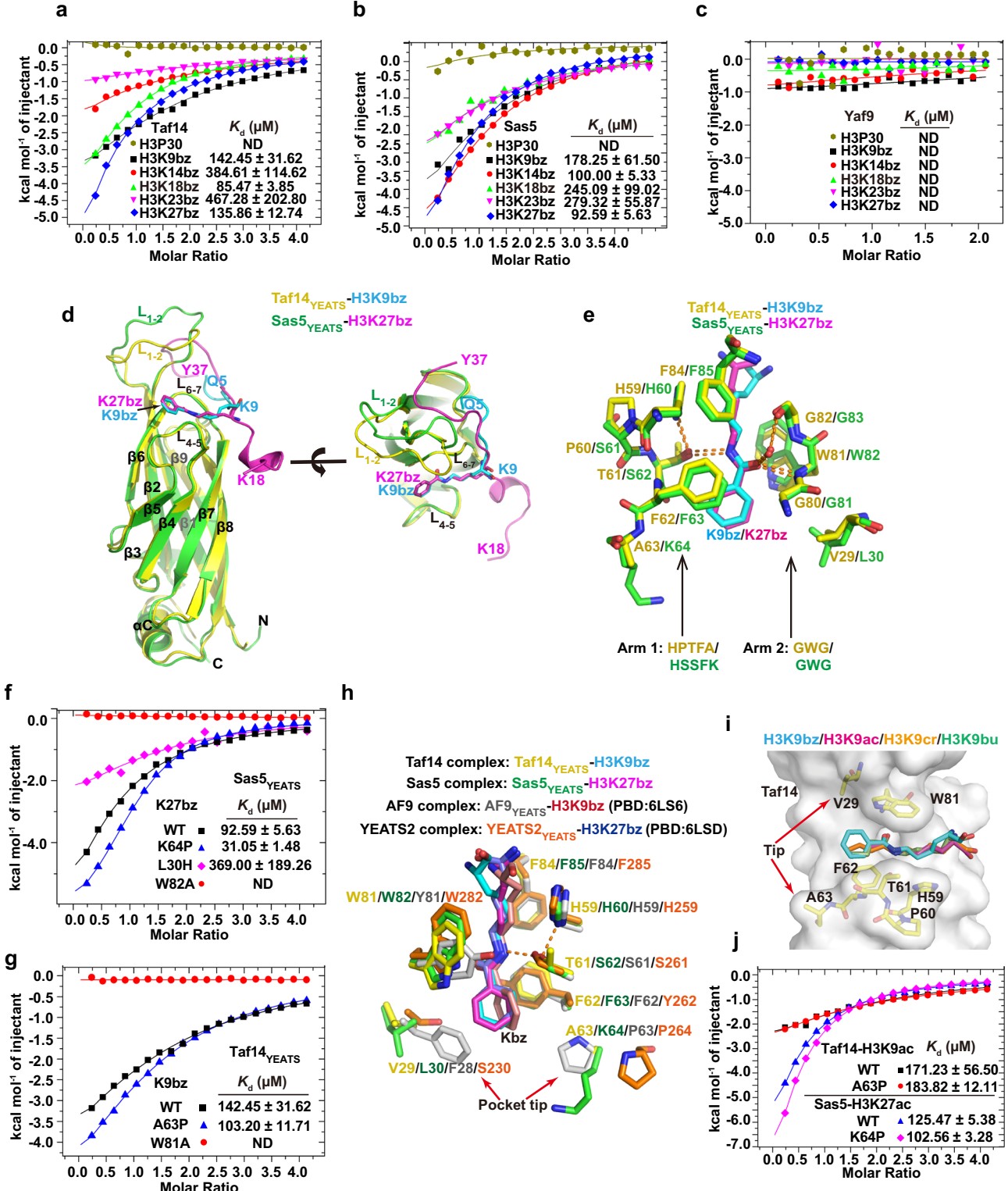

K64P mutant, which may generate a more closed binding pocket with increased hydrophobicity, substantially strengthened the interaction between Sas5_YEATS and H3K27bz but only slightly increased the binding with H3K27ac (Fig. 5f, j). Similarly, the Taf14 A63P mutant increased the interaction with K9bz by 1.4-fold but did not affect the binding with K9ac (Fig. 5g, j). A similar tip-sensing mechanism is also reported in human AF9 and YEATS2 proteins[6]. Collectively, our results reveal that YEATS domains from Taf14 and Sas5 function as readers for Kbz and

utilize the conserved molecular mechanism for Kbz recognition that is shared by YEATS domains from mammalian counterparts AF9 and YEATS2 proteins.

**The Bromodomain from Sth1 has Kbz binding activity.** Previous studies demonstrated that bromodomains could not bind benzoylated lysine[6,35]. Here, we re-examined the possibility of whether yeast bromodomains could bind benzoylation marks. There are 15 bromodomains in *S. cerevisiae*, and we purified four

**Fig. 5 YEATS domains from Taf14 and Sas5 are Kbz readers. a** ITC analyses show that Taf14$_{YEATS}$ has different binding affinities with Kbz peptides. **b** ITC analyses show that Sas5$_{YEATS}$ could bind Kbz peptides with different binding affinities. **c** ITC analyses show that Yaf9$_{YEATS}$ does not bind any Kbz peptides tested or that the binding affinity is beyond the detection limit. **d** Superimposition of Taf14$_{YEATS}$-H3K9bz and Sas5$_{YEATS}$-H3K27bz structures. The Taf14$_{YEATS}$ and Sas5$_{YEATS}$ are shown in yellow and green; H3K9bz and H3K27bz peptides are shown in blue and magenta. **e** The benzoyl groups in H3K9bz and H3K27bz peptides are inserted into a conserved binding pocket sharing nearly identical interacting networks. The residues involved in hydrophobic contacts and hydrogen bonds (orange dashed lines) are localized in two arms, embracing the benzoyl group. The red dot is the water molecule. **f** ITC results show the effects of Sas5$_{YEATS}$ mutations on the interaction between Sas5$_{YEATS}$ and H3K27bz peptide. **g** ITC results show the effects of Taf14$_{YEATS}$ mutations on the interaction between Taf14$_{YEATS}$ and H3K9bz peptide. **h** Superimposition of four YEATS-Kbz complex structures, including AF9-H3K9bz (gray-red), YEATS2-H3K27bz (orange-navy blue), Taf14-H3K9bz (yellow-cyan), and Sas5-H3K27bz (green-magenta). The residues at the tip regions of the benzoyl-binding pocket are different and indicated by arrows. **i** The acyl-binding pocket of Taf14$_{YEATS}$. Four Taf14$_{YEATS}$ structures are superimposed, including Taf14-H3K9bz, Taf14-H3K9ac (PDB: 5D7E), Taf14-H3K9cr (PDB: 5IOK), and Taf14-H3K9bu (PDB: 6MIQ). The benzoyl, acetyl, crotonyl, and butyryl groups are shown in cyan, magenta, orange, and green, respectively. The tip residues of V29 and A63 are involved in the fine-tuning of pocket shapes to accommodate different acylation marks. **j** ITC results reveal that the mutations of tip residues on Taf14$_{YEAST}$ and Sas5$_{YEATS}$ have negligible effects on binding with acetylated peptides.

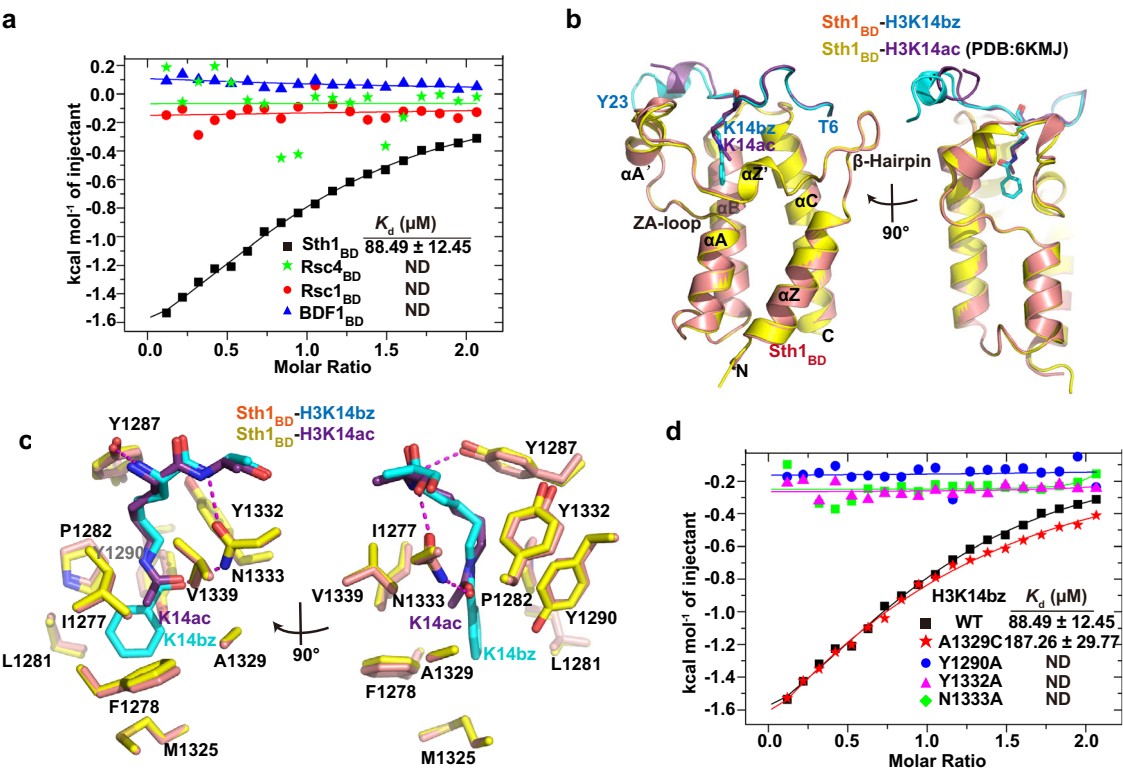

**Fig. 6 Sth1 bromodomain has Kbz binding activity. a** ITC analyses of four yeast bromodomains binding with H3K14bz peptides. Sth1$_{BD}$ exhibits a detectable interaction with the H3K14bz peptide ($K_d = 89$ μM). **b** Superimposition of Sth1$_{BD}$-H3K14bz (Sth1:orange; H3, cyan) and Sth1$_{BD}$-H3K14ac (Sth1: yellow; H3, purple) complexes shows almost identical conformations of Sth1$_{BD}$ and H3 peptide, except for the C-terminal 3$_{10}$-helix of H3. **c** Detailed interaction networks between acylated lysine and Sth1$_{BD}$. Benzoyl and acetyl groups are shown in cyan and purple, respectively. Hydrophobic contacts and hydrogen-bonding networks are shown. **d** ITC assays showing the effects of Sth1 mutations on the interaction between Sth1$_{BD}$ and H3K14bz peptide.

of these bromodomains and characterized their binding abilities with H3K14bz and H3K27bz peptides (Fig. 6a and Supplementary Fig. 6a). Although most bromodomains showed negligible H3K14bz or H3K27bz binding abilities, the Bromodomain from Sth1 (Sth1$_{BD}$), the ATPase subunit in the Remodel the Structure of Chromatin (RSC) complex, surprisingly exhibited robust interaction with H3K14bz ($K_d = 89$ μM), albeit weaker than that with H3K14ac ($K_d = 16$ μM)[36] (Fig. 6a)

To reveal how Sth1$_{BD}$ recognizes Kbz, we determined the crystal structure of Sth1$_{BD}$ in the complex with the H3K14bz peptide at a resolution of 1.4 Å (Fig. 6b and Table 1). The H3 peptide is well ordered, as evidenced by good electron density from T6 to Y23 (Supplementary Fig. 6b). Histone H3 sits on the acidic apical region of the Sth1$_{BD}$ helical bundle and makes extensive contacts with the ZA-loop, BC-loop, αZ′, αA′, αB, and

αC of Sth1$_{BD}$ (Fig. 6b and Supplementary Fig. 6c, d). The overall complex conformation is highly similar to that of Sth1$_{BD}$-H3K14ac, except for the shift of the H3 C-terminal 3$_{10}$-helix (K18-A21) (Fig. 6b). In both K14bz-bound and K14ac-bound[36] Sth1$_{BD}$ complexes, the acylated lysines are inserted into an identical binding pocket sharing the same set of hydrophobic residues (Y1287, Y1332, Y1290, V1339, I1277, F1278) and hydrogen-bonding networks mediated by N1333 and Y1287 (Fig. 6c). Mutations of these hydrophobic residues and hydrogen-bonding residues completely disrupted the interactions between Sth1$_{BD}$ and K14bz-containing peptides, validating the importance of these residues in recognizing acylated lysines (Fig. 6d). The indistinguishable configurations of the Kbz- and Kac-binding pockets indicate that the acylation-binding pocket of Sth1$_{BD}$ intrinsically has adequate space to accommodate the large

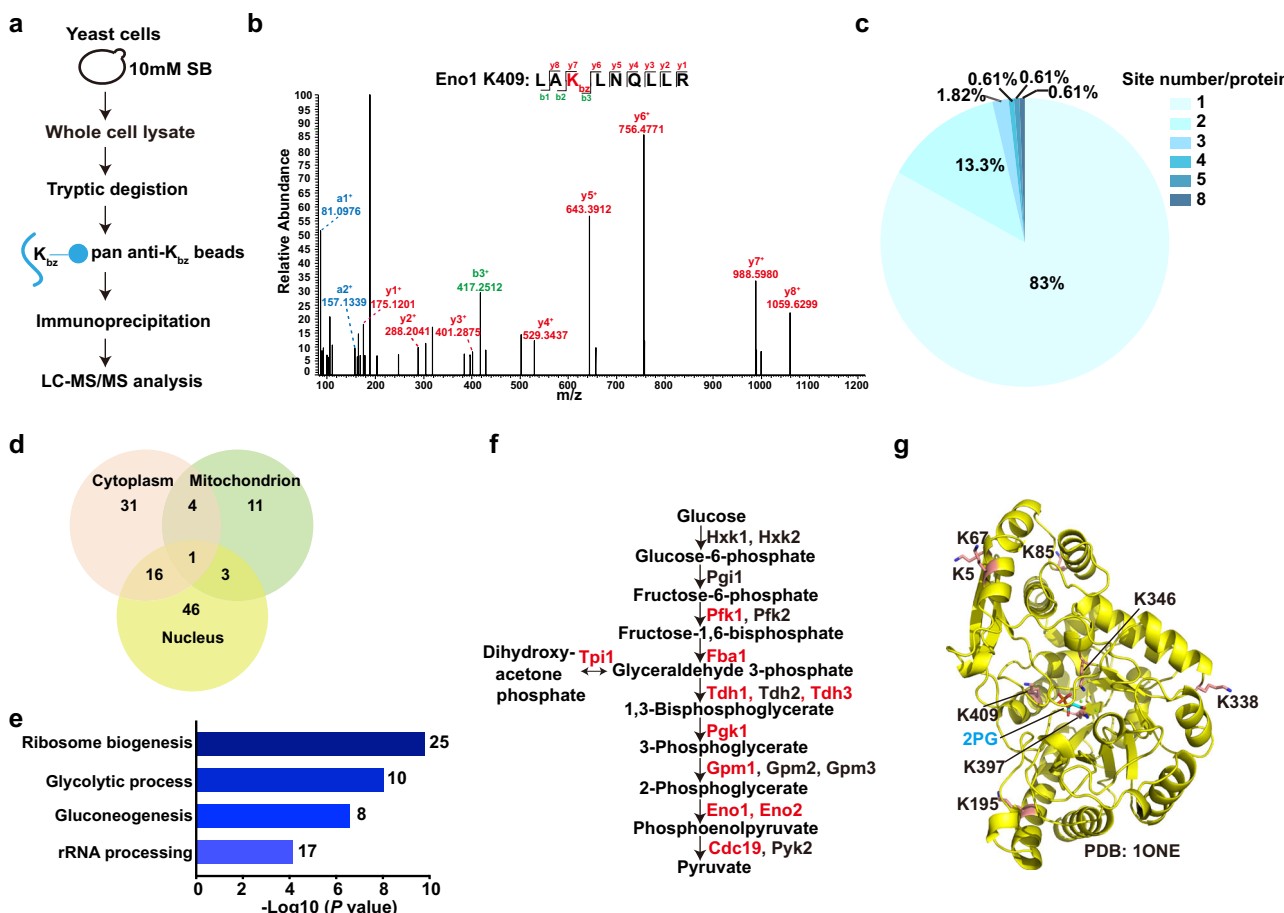

**Fig. 7 Proteome-wide screening of Kbz-modified proteins in yeast cells. a** The workflow used for HPLC–MS/MS-based non-histone Kbz sites identification in *S. cerevisiae*. **b** The MS/MS spectrum of an Eno1 peptide (LAK$_{+104.0268}$LNQLLR) harboring one benzoylated site. Predicted y- and b-type ions are listed above and below the peptide sequence, respectively. Matched ions are labeled in the spectra. **c** Distribution of non-histone Kbz sites based on the site number per protein. **d** Venn diagram showing the subcellular compartment distribution of non-histone Kbz proteins. **e** Gene ontology analysis associated with significant regulated genes (*P* < 0.05) of non-histone Kbz proteins. *P* values are derived from one-sided Fisher's exact test. **f** Overview of glycolysis pathway. The enzymes with identified benzoylation sites are shown in red. **g** Eight benzoylation sites on Eno1 are shown in the structural model of Eno1 (PDB: 1ONE). The Eno1 substrate, 2-phosphoglycerate, is shown in cyan. K346, K397, and K409 are adjacent to the substrate-binding pocket.

benzene ring without any structural rearrangement. Reconciling the recent finding that the human bromodomain (BRD9, CECR2, and TAF1$_{BD2}$) can bind butyrylated or crotonylated lysine[35], we conclude that the plasticity of bromodomains could render a subset of bromodomain capable of binding diverse acylation longer than acetyl marks.

**Proteome-wide screening of non-histone Kbz sites in yeast cells.** Immunoblotting of whole-cell lysate with the anti-Kbz antibody indicated that some non-histone proteins were also benzoylated in yeast cells, even in the absence of sodium benzoate (Supplementary Fig 7a). To globally identify non-histone protein substrates bearing Kbz and their modification sites, we carried out proteomic screening involving extraction of whole yeast cell lysate proteins, tryptic digestion, affinity enrichment of Kbz peptides with pan anti-Kbz antibody-conjugated agarose beads, and HPLC-MS/MS analysis (Fig. 7a). We identified a total of 207 Kbz sites on 149 non-histone proteins (Supplementary Data 2). Three representative MS/MS spectra of non-histone Kbz peptides are shown in Fig. 7b and Supplementary Fig. 7b, c. Further analysis revealed that 83% of these proteins were modified at only one Kbz site (Fig. 7c).

To explore the subcellular localization of non-histone Kbz proteins, we carried out a cellular compartment analysis of the

non-histone Kbz proteins. Approximately 13.9% of non-histone Kbz proteins were enriched in the mitochondrion, and 48.2% and 40.0% of non-histone Kbz proteins were localized in the nucleus and cytoplasm respectively (Fig. 7d). Gene ontology enrichment analysis indicated that most non-histone Kbz proteins are involved in ribosome biogenesis ($P = 1.58 \times 10^{-10}$), glycolytic process ($P = 8.81 \times 10^{-9}$), gluconeogenesis ($P = 2.63 \times 10^{-7}$), and rRNA processing ($P = 7.32 \times 10^{-5}$), suggesting that the benzoylation modification may have broad functions in both the nucleus and cytoplasm (Fig. 7e and Supplementary Data 3).

Glycolysis is the multiple-step process of releasing energy that converts glucose to pyruvate. Remarkably, 10 out of the 18 enzymes involved in glycolysis contain benzoylation modifications (Fig. 7f). Of these, Eno1 was heavily modified with eight benzoylation sites (Fig. 7g). Eno1 is a phosphopyruvate hydratase that catalyzes the conversion of 2-phosphoglycerate to phosphoenolpyruvate during glycolysis. Interestingly, three benzoylation sites (K346, K397, and K409) are located close to the active site of Eno1 (2-phosphoglycerate-binding site in Fig. 7g). Previous studies showed that mutation of K346 to Ala reduced $k_{cat}$ over 100,000-fold and abrogated the initial rate of the enzymatic reaction[37,38]. It is expected that benzoylation of these sites (K346, K397, and K409) may abolish the enzymatic activity of Eno1. Therefore, these results indicate that lysine benzoylation

is a widespread modification in yeast cells and implicate a potential link between lysine benzoylation modification and cell metabolic pathways.

## Discussion

Advances in high-resolution mass spectrometry technology and instrumentation have allowed us to identify novel histone PTMs with unprecedented efficiency, greatly deepening our understanding of the epigenome[28]. Histone lysine benzoylation (Kbz) is one of the new members added to the growing list of histone modifications[5]. Previous studies have verified the existence of histone benzoylation in human, mouse, and Drosophila cells[5]. Identifying histone benzoylation sites in yeast cells confirms that histone benzoylation is an evolutionarily conserved modification in eukaryotes. We identified 9 Kbz sites on core histones even without sodium benzoate treatment, indicating the inherent occurrence of histone benzoylation in yeast cells. Notably, yeast histones have three Kbz-specific sites (H3K42, H3K115, and H4K44) with no other acylation modification identified. Moreover, these three unique Kbz sites are also yeast-specific sites that do not exist in mammalian cells, and all are localized in the globular domains of H3 and H4, suggesting certain functional plasticity of histone benzoylation in different species. Inspection of the nucleosome structure reveals that these sites are close to the DNA-contacting interfaces. It is reminiscent of a previous finding that the lysine glutarylation of H4K91, localizing in the interface between H4 and H2B, decreases nucleosome stability and influences chromatin structure[4]. We hypothesize that the Kbz modifications in H3/H4 globular domains will also affect nucleosome/chromatin stability and render transcription activation, which merits further investigation.

The identification of writers, erasers, and readers for a specific lysine acylation mark is critical to study its physiological functions[11,29,39,40]. Which enzyme is responsible for depositing histone benzoylation in mammalian cells remains unknown[5]. We revealed that the yeast Gcn5 complex could act as a histone benzoyltransferase. Therefore, it could be expected that the GCN5-containing complex may also function as a histone benzoyltransferase in mammalian cells. One previous study revealed that the human GCN5 complex has a histone succinyltransferase activity, and the succinyl-CoA fits into the active pocket of hGCN5[24]. Since the succinyl group and benzoyl group have similar dimensions (Supplementary Fig. 3i), the active site of GCN5 has the adequate space to accommodate the benzoyl group without dramatic structural rearrangement, which was confirmed by the comparable benzoyl-CoA and succinyl-CoA binding abilities of the Gcn5-Ada2 complex (Supplementary Fig. 3j). It is noteworthy that our results do not contradict the previous finding that hGcn5 displays nearly no activity toward more extended acylation marks, such as Kbu and Kcr[41]. Consistent with previous findings that the binding of Ada2 to Gcn5 stimulates Gcn5′s HAT activity[25], the presence of Ada2 increased Gcn5′s benzoyltransferase activity as well. Moreover, the complete SAGA complex had higher catalytic activity than the Ada2-Gcn5 subcomplex (Supplementary Fig. 3c, d). This suggests that the ability of Gcn5 to catalyze long acylation reactions are facilitated by other subunits of the Gcn5 complex. The binding partners in the HAT complex may fine-tune the activity pocket of Gcn5 to accommodate diverse acyl-CoA with different volumes and polarities, thus generating an enzyme with multiple acylation activities. This multifaceted enzyme complex may help cells acclimatize to divergent metabolic states and physiological circumstances.

Histone deacetylases (HDACs) can act as erasers to catalyze the removal of acylation marks. Class III sirtuin-family HDAC proteins can erase various acylation marks other than acetylation,

including crotonyl[27], malonyl[42], succinyl[42], and glutaryl[4] marks. A previous study showed that mammalian SIRT2, but not other HDACs, could remove Kbz from a synthetic peptide (PEPTKbzSAPAPK) in vitro[5]. Here we also showed that Hst2 in yeast has debenzoylase activity in vivo and in vitro, confirming that class III sirtuin family HDACs are the major debenzoylases across species. Biochemical and structural studies of Hst2 manifest the substrate specificity of Hst2 with a preference for RK motifs. Notably, the Hst2 residues involved in histone binding are not well conserved in class III HDACs (Supplementary Fig. 4k), indicating that other HDACs might have distinct substrate specificities. For example, human SIRT3, as a histone de-β-hydroxybutyrylase, has a unique substrate preference and is incapable of removing Kbhb at sites with flanking glycine residues[43]. Considering the possible diverse substrate specificities of different HDACs, we cannot rule out the possibility that, in addition to SIRT2, other mammalian SIRT-family proteins might also function as histone debenzoylases by recognizing other sites distinct from the previously tested peptides.

Our structural studies of Hst2 complexes reveal the molecular mechanism by which Hst2 recognizes Kbz. Hst2 contains a sealed hydrophobic pocket composed of conserved nonpolar residues. Four residues (F67, I117, I181, and F183) at the tip of the acyl-binding channel generate geometry constraints on which acylation marks could fit. The subtle structural rearrangement of the tip residues I117 and F67 enables snug fitting of the benzoyl group (Fig. 4h). We speculate that these tip residues are the major determinants for which modification could be catalyzed by HDACs. Mutations of these residues altered the catalytic efficiencies of Hst2 toward different acylations (Fig. 4i, j). Thus, these results offer an opportunity to engineer a versatile HDAC to a more specific deacylase through mutations of the tip residues. Since many HDACs can promiscuously remove multiple types of acylation marks[11,42,44,45], it is still challenging to study the function of a specific modification erased by HDACs. The generation of a more specific deacylase will assist the functional dissection of a particular acylation mark regulated by HDACs.

Our biochemical and structural studies confirm the previous conclusion that YEATS domains could function as readers for histone Kbz through a well-conserved recognition mechanism shared by Taf14, Sas5, AF9, and YEATS2[6]. In addition to YEATS domains, we illustrate that the bromodomain from Sth1 could bind H3K14bz with a higher affinity than YEATS domains from Taf14 or Sas5. This finding is unexpected because two previous studies demonstrate that bromodomains cannot bind benzoylated lysine[6,35]. The Sth1$_{BD}$-H3K14bz structure clearly shows that the acylation-binding pocket of Sth1 has adequate space to accommodate the large benzene ring without any structural rearrangement of Sth1$_{BD}$ (Fig. 6c). This indicates that although most bromodomains have small acyl-binding pockets suitable for short acetyl marks, certain bromodomains can bind diverse acylation marks other than acetylation[35]. Thus, we need to reassess the role of ubiquitous bromodomains in recognizing other acylation marks and the associated non-acetylation reader functions in the future.

Mounting evidence suggests that lysine acylations, such as Kbhb, Khib, Kmal, and Ksucc, are widely distributed in non-histone proteins[10,11,16,17]. We report the first proteomic profile of the global lysine benzoylation in yeast. These benzoylated proteins are enriched in the ribosome pathway and metabolic processes, implicating potential roles of Kbz in these cellular processes. The identification of proteins containing Kbz is a critical step toward the understanding of the diverse functions of Kbz, including but not limited to transcriptional regulation, metabolic response, and ribosome biogenesis. Moreover, further investigations on the functional changes of these benzoylated

proteins may shed light on the molecular mechanism of anti-fungal activity of the widely used preservative sodium benzoate and help develop a new antifungal strategy.

## Methods

**Antibodies**. Anti-Kbz (PTM Biolabs Inc., PTM-762, 1:1000, Lot# 1314381619I926), anti-Kac (Cell signal, 9441 S, 1:1000, Lot# 14), anti-H3N (Active motif, 39763, 1:700, Lot# 20418023), anti-H4 (Active motif, 61521, 1:3000, Lot# 31416005), and anti-tubulin (Sigma, T6199, 1:3000, Lot# 048M4751V) antibodies were used in this study. The HRP secondary antibody and the fluorescent secondary antibodies were used: Peroxidase AffiniPure Goat Anti-Mouse IgG (Jackson ImmunoResearch, 1:5000, 115-035-003), Peroxidase AffiniPure Goat Anti-Rabbit IgG (Jackson ImmunoResearch, 1:5000, 111-035-003), IRDye 800CW goat anti-rabbit IgG (LI-COR P/N 923-32211, 1:5000, Lot# D00304-15) and IRDye 680RD goat anti-Mouse IgG (LI-COR P/N 926-68070, 1:10000, Lot# D01014-04). The fluorescent western blots were detected on Odyssey CLx (LI-COR) and quantified by Image Studio (ver. 5.2.5).

**Plasmids and Strains**. *Saccharomyces cerevisiae* Hst2$_{FL}$ (residues 1-357), Hst2 (residues 8-294), Taf14$_{YEATS}$ (residues 2-137), Sas5$_{YEATS}$ (residues 2-139), and Sth1$_{BD}$ (residues 1248-1359) were amplified by high-fidelity *FastPfu* Fly DNA polymerase (Transgen Biotech, China) (Supplementary Table 1), and then sub-cloned into a modified pET28b vector with an N-terminal 6*His-Sumo tag by a ClonExpress II One Step Cloning Kit (Vazyme, China). The Ada2 (residues 1-120) and Gcn5 (residues 67-317) fragments were cloned into a modified pETDuet4 vector with an N-terminal 6*His tag by a ClonExpress MultiS One Step Cloning Kit (Vazyme, China).

Yeast strains are in the BY4742 background (*MATα his3Δ1 leu2Δ0 lys2Δ0 ura3Δ0*). The HAT and HDAC deletions in this paper were from MATalpha Library, which generated by a PCR-based gene deletion strategy to generate start to stop codon deletions. Each deletion was replaced with a KanMX module and strains were grown in YPD with 200 µg/ml G418 (Sangon Biotech, A600958-0001) (http:// sequence-www.stanford.edu/group/yeast_deletion_project/deletion3.html). For SAGA complex purification, the *SPT20* gene was tagged by a C-terminal TAP tag[46,47]. The temperature-sensitive mutant *esa1-531* strain was a gift from Junbiao Dai[3,10].

**Protein expression and purification**. The plasmids were transformed into *Escherichia coli* Rosetta cells. Proteins were overexpressed with induction by 0.2 mM isopropyl β-D-1-thiogalactopyranoside (IPTG) for 16–18 h at 18 °C. Harvested cells were resuspended in lysis buffer containing 50 mM Tris-HCl, pH 8.0, 400 mM NaCl, 10% glycerol, 2 mM 2-mercaptoethanol, and protease inhibitor cocktail. The cells were broken by sonication on ice and then centrifuged at 20,000 g for 40 min. The supernatant was incubated with Ni-NTA beads (Qiagen, USA) for 2 h at 4 °C, and the tag-free proteins were then eluted by on-beads digestion with ULP1 protease (for pET28b-Sumo vectors) or 3 C protease (for pETDuet4 vectors) added at a molar ratio of 1:200. Eluted proteins were further purified by size-exclusion chromatography (SEC) on Hiload Superdex 75 or Hiload Superdex 200 columns in the buffer of 25 mM Tris-HCl, pH 8.0, and 150 mM NaCl. Purified proteins were concentrated and stored at −80 °C. All mutations were introduced by PCR-based site-directed mutagenesis, and the presence of appropriate mutations was confirmed by DNA sequencing. Mutant proteins were purified by using the same protocol as described above.

**Purification of SAGA complex**. The C-terminus of the Spt20 subunit of the SAGA complex was tagged with a TAP tag. The SAGA complex was purified using a standard TAP procedure as described previously[46]. Typically, 10 L of yeast cells were harvested and broken by SPEX 6857 Freezer Mill (SPEXSamplePrep, USA). After centrifugation at 200,000 g for 1 h at 4 °C, the supernatant was incubated with 800 µL prewashed IgG-Sepharose beads (GE Healthcare, 17096901, USA) for 4 h at 4 °C. The protein sample was released from the beads by TEV protease digestion for 4 h. The eluted protein complex was concentrated to 40–50 µL for in vitro acetylation and benzoylation assays.

**Histone extraction from yeast**. Typically, 100 mg yeast cell pellets were harvested and suspended in 1 ml buffer A (1 M sorbitol, 50 mM Tris-HCl, pH 7.5, 5 mM MgCl$_2$, 5 mM 2-mercaptoethanol) and centrifuged for 5 min at 900 g. Then, the pellet was resuspended in 1 ml buffer A with 60 µl zymolyase. After incubation at 35 °C for 30 min, the pellet was resuspended in 700 µl buffer B (1 M sorbitol, 50 mM MES, pH 6.0, 5 mM MgCl$_2$) and incubated on ice for 5 min. Harvested pellets were resuspended in the buffer containing 50 mM MES, pH 6.0, 75 mM KCl, 0.5 mM CaCl$_2$, 0.1% NP-40, 0.5% IGEPAL CA-630. After incubation on ice for 5 min, the pellet was washed with 700 µl buffer C (50 mM MES, pH 6.0, 75 mM KCl, 0.5 mM CaCl$_2$, 0.1% NP-40). Harvested pellet was resuspended in 120 µl of 0.25 M HCl and rotated at 4 °C for 2 h. The histone samples were then precipitated by adding acetone at a ratio of 1:8. The white pellet appeared and was washed with 400 µl acidified acetone. Finally, the pellet was resuspended in 10 µl-20 µl of water. The concentration of histones was measured by a BCA Protein Assay Kit

(TIANGEN, PA115-01, China). The samples were analyzed by western blot or by mass spectrometry.

**Preparation of whole-cell extract from yeast**. After harvesting the yeast cells, 100% TCA was added to 5 ml yeast cells to a final concentration of 10% (v/v) to stop all enzyme activities in the cells. The sample was incubated on ice for at least 10 min. The pellet was resuspended in 500 µL ddH$_2$O, and 500 µL 0.2 M NaOH. After incubation for 10–30 min at room temperature, the cell pellet was lysed in lysis buffer containing 9 M urea, 20 mM HEPES, pH 8.0, 2.5 mM sodium pyrophosphate, and 1 mM sodium metavanadate for 0.5 h, and then broken by sonication on ice. After centrifugation at 20,000 g for 15 min at 4 °C, harvested protein was measured by a Pierce$^{TM}$ BCA Protein Assay Kit (Thermo, 23227, USA) and analyzed by western blot.

**Chemical propionylation and trypsin digestion**. Chemical propionylation was performed using a standard procedure as described previously[5]. Whole-cell proteins were dissolved in 250 µL of 0.1 M NH$_4$HCO$_3$, pH 8.0, followed by the addition of 3.0 µL propionic anhydride. The sample was adjusted to pH 8–9 and incubated for 1 h at room temperature. After adding another 3.0 µL of propionic anhydride and keeping the pH at 8–9 by the addition of 1 M NaOH, the mixture was incubated for another 1 h, and the reaction was blocked by adding 3.0 µL of 2-aminoethanol. Then, the sample was incubated at room temperature for 15 min.

**Reductive alkylation and trypsin digestion**. Whole-cell proteins and histone samples were reduced by adding 5 mM TCEP and incubated for 30 min at room temperature. Then, the samples were alkylated by adding 10 mM iodacetamide. After incubating in the dark for 30 min, the samples were digested with trypsin (protein: trypsin = 100:1) at pH 8–9 and 37 °C overnight. We have compared the data quality from samples processed by two different ways: one with two rounds of propionylation and another with reductive alkylation without any propionylation. The results showed that the quality of the spectra from reductive alkylation was better than that with two rounds of propionylation. Thus, the histone benzoylation data presented in this manuscript are all from the samples prepared by reductive alkylation.

**Immunoprecipitation enrichment**. The enrichment of Kbz peptides was carried out as described previously. Briefly, the peptides were incubated with pre-balanced pan anti-Kbz antibody-conjugated beads or pan-Kbz antibody at 4 °C with gentle shaking. For the latter, the samples were incubated with the ProA/G magnetic beads for another 4 h. Then, the beads were washed three times with 500 µL wash buffer containing 50 mM Tris-HCl, pH 8.0, 100 mM NaCl, 1 mM EDTA, and 0.5% NP-40, followed by washing twice with 200 µL buffer containing 50 mM Tris-HCl, pH 8.0, 100 mM NaCl, 1 mM EDTA, and once with water. The bound peptides were eluted twice with 0.15% trifluoroacetate (TFA). After desalting with C18 monoSpin column (GL Science, Japan), the combined samples were vacuum-dried.

**HPLC-MS/MS analysis**. Peptide samples were dissolved in HPLC buffer A (0.1% formic acid in water, v/v) and loaded onto a capillary C18 column (30 cm length × 75 µm I.D., 1.9 µm particle size, Dr. Maisch GmbH, Ammerbuch, Germany), which was connected to the EASY-nLC 1200 UHPLC system (ThermoFisher Scientific, San Jose, CA) with a column heater (SCO-2110 pro, China Innovation Instrument Co., Ltd). Peptides were separated with a gradient of 2–100% HPLC buffer B (0.1% formic acid in 80% acetonitrile, v/v) at a flow rate of 300 nL/min over 180 min. The eluted peptides were analyzed by a Q Exactive mass spectrometer (ThermoFisher Scientific, San Jose, CA) using a nanospray source. Full mass scans were acquired in the *m/z* range of 300−1800 with a mass resolution of 70,000 at *m/z* 200. The 20 most intensive ions were fragmented with 28% normalized collision energy, and tandem mass spectra were acquired with a mass resolution of 17,500 at *m/z* 200. Column temperature was set as 55 °C.

**Protein sequence database searching**. The benzoylation identification search was performed with the PEAKS Studio 8.5 (ThermoFisher, USA) against the Uniprot Yeast protein database (6721 entries, http://www.uniprot.org). The following parameters were used during sequence alignment: methionine oxidation, protein N-terminal acetylation, lysine acetylation, lysine mono-/di-/tri-methylation, arginine mono-/di-methylation, and lysine benzoylation were specified as variable modifications. Cysteine carbamidomethylation was set as the fix modification when alkylation was performed. Peptide N-terminal and lysine propionyl were set as variable modifications when propionylation was performed. The maximum missing cleavage was set at 4, the maximum variable PTM number per peptide was set at 6, and the mass tolerance was set at 20 ppm for precursor ions and ±0.02 Da for MS/MS. Trypsin/P was specified as a cleavage enzyme allowing up to four missing cleavages. FDR thresholds for peptides and modification sites were specified at 1%. For histone samples, Kbz sites with Ascore ≥ 20 were chosen for further analysis. For whole-cell lysate analysis, Kbz sites with Ascore ≥ 13 were used.

**HPLC-MS/MS analysis of benzoyl-CoA**. The BY4742 cells were grown in YPD to log phase, and then 10 mM sodium benzoate was added to the medium for 6 h. The yeast cells (10 OD) were collected and centrifuged for 3 min at room temperature. Cells were resuspended in 1 ml of prechilled quenching buffer containing acetonitrile/methanol/$H_2O$ = 2:2:1 (v/v/v) and transferred into 2 ml tubes. After filling 500 μL glass beads, the cells were homogenized 10 times for 10 s with 2 min intervals between each homogenization pulse (BEAD RUPTOR, OMNI International, USA). A hole was poked into the tube bottom using a hot needle in order to separate the cell lysate from the beads. The supernatant was collected and centrifuged for at least 20 min at 17,000 $g$ at 4 °C to remove any remaining cell debris, and then dried out in a SpeedVac. The samples were resolved in 50 mM $NH_4OAc$, pH 7.0 right immediately before analysis. The LC-MS/MS method was described as previously reported with a slight modification[48]. In short, the samples were loaded onto a BEH amide column (100 × 2.1 mm i.d., 3.5 μm; Waters, USA) by the HPLC with an Agilent 1290 autosampler (Agilent, USA) coupled to a QTRAP 6500 (Sciex, USA) at 25 °C. Mobile phase A was 10 mM ammonium acetate and 15 mM ammonium hydroxide in water with 3% acetonitrile, pH 9.0, and mobile phase B was acetonitrile. The linear gradient was used as follows: 0 min, 85% B; 3 min, 85% B, 10 min, 35% B; 15 min, 35% B, 15.1 min, 85% B, 20 min, 85% B. The flow rate was set at 0.6 mL/min. MRM mode was used for quantitative analysis. The source parameter was set as follows: Positive mode; Curtain gas: 35; Collision Gas: Medium; Ion Spray Voltage: 5500; Temperature: 500; Ion source Gas1:50; Ion source Gas2:60; the collision Energy was set at 46. Dwell times of 100 ms were used for all transitions, and cycle times were set to 0.5 s. The acquired data were extracted and quantified by Skyline v21 (University of Washington, MacCoss Lab, https://skyline.ms) under small molecule mode. Briefly, the MRM transitions for each sample were integrated to generate ion current peak areas representing each of the fragment ion signals. The peak area for each sample was calculated for quantitation, and skyline data were directly exported as a text file for further statistical analyses. The standard curve was obtained by varying concentrations of benzoyl-CoA standards (0.01, 0.03, 0.05, 0.10, 0.20, 0.50, 1.00, 2.50 μM). The linear fitting between corresponding peak areas and benzoyl-CoA concentrations was achieved.

**Immunoblotting-based acetyltransferase and benzoyltransferase assays**. The reactions were performed in 50 mM Tris-HCl, pH 7.5, 1 mM DTT, 3.2 mM $MgCl_2$, 40 mM NaCl, 0.1 mM EDTA, 2 μM *Xenopus laevis* histone octamer, and the 5 μM Ada2-Gcn5 or 10 μL SAGA complex. After incubation for 5 min at 30 °C, 100 μM (for Fig. 2b, c) or 250 μM (for Supplementray Fig. 3c, d) acetyl-CoA/benzoyl-CoA was added to the mixtures to start the reactions. The reactions were quenched at 1 h, 1.5 h, and 2 h by the addition of trifluoroacetate (TFA) to 0.5%. The levels of acetylation and benzoylation were assessed by western blot using pan-anti-Kac and pan-anti-Kbz antibodies.

**Mass spectrometry-based benzoyltransferase assays**. The reactions were performed in 50 mM Tris-HCl, pH 7.5, 1 mM DTT, 3.2 mM $MgCl_2$, 40 mM NaCl, 0.1 mM EDTA, 2 μM *Xenopus laevis* histone octamer and the 5 μM Ada2-Gcn5. After incubation for 5 min at 30 °C, 100 μM acetyl-CoA/benzoyl-CoA (Sigma, A2181, B1638, USA) was added to the mixtures to start the reactions. The reactions were quenched at 2 and 2.5 h by the addition of trifluoroacetate (TFA) to 0.5%. The products were precipitated by directly adding 6 volumes of cold acetone and incubated at -20 °C for overnight. The next day, the pellets were collected and resuspended in 50 μl of freshly prepared 8 M urea. Then, the samples were aliquoted into four parts and digested with trypsin (protein: trypsin=1:50) at pH 8-9 for 0.5 h, 2 h, 4 h, and overnight. After digestion, formic acid was added to a final concentration of 0.1% and frozen at -80 °C immediately. The samples were combined and desalted by a C18 mono-spin column (GL Science, Japan) and dried out by a SpeedVac. Finally, the samples were reconstituted by 0.1% formic acid and analyzed by the LC-MS/MS. Non-enzymatically reaction were performed under the same conditions in the absence of Gcn5-Ada2.

**The DNTB acylation activity assay**. Kinetic measurements comparing rates of acetylation and benzoylation were performed using the 5,5′-dithiobis-(2-bitrobenzoic acid) (DTNB, D218200-1G, Sigma) assay[41]. Reactions contained 2 μM Gcn5-Ada2, 250 μM H3 peptide amino acids 1-18 (ARTKQTARKSTGGK APRKY), 100 mM HEPES pH7.8, 50 mM NaCl and 500 μM acetyl-CoA or benzoyl-CoA. Each reaction was incubated for 5 min at 30 °C before adding acyl-CoA to start the reaction. The reaction was quenched at the indicated time points by adding two volumes of freshly prepared quenching buffer (3.2 M guanidine-HCl, 100 mM sodium phosphate pH 6.8). The samples were added to one volume of 4 mM DTNB dissolved in 100 mM sodium phosphate (pH 6.8) and transferred to a 384-well polystyrene clear-bottom plate (CORNING, 3680), and the absorbance at 412 nm was measured in a Bio-TEK plate reader. Absorbances were converted to concentrations using a standard curve generated by HSCOA standards. The acetylation reaction was quenched after 0.5, 1.0, 1.5, 2.0, and 2.5 min, and the benzoylation reaction was quenched after 1, 2, 3, and 4 h. All acylation rates were corrected by subtracting the rate of acyl-CoA consumption by Gcn5-Ada2 in the absence of peptide. The acetylation assays were performed in triplicate, but the benzoylation assay was not repeated due to the limitation of materials.

**MALDI-TOF-Based debenzoylase and deacetylase assay**. A MALDI-TOF/TOF mass spectrometry-based debenzoylase and deacetylase assay was performed to detect molecular weight decreases of 104 Da and 42 Da, respectively. For the debenzoylase assay, the substrate peptides, including H3K9bz, H3K14bz, H3K18bz, H3K23bz, and H3K27bz, were chemically synthesized by Scilight-Peptide, China (Supplementary Table 2). For the deacetylase assay, the substrate peptides of H3K9ac and H3K14ac were synthesized by Genscript, China. The reactions were performed in 50 mM Tris-HCl, pH 7.5, 1 mM DTT, 3.2 mM $MgCl_2$, 1 mM $NAD^+$, and 10 μM peptide at room temperature. The final concentrations of Hst2[FL], Hst2[D68A/D70A], Hst2[H135A], Hst2[V182A], Hst2[F184A], Hst2[I117F], and Hst2[I117V] proteins were 2 μM for the debenzoylase assay and 0.5 μM for the deacetylase assay, respectively. The reactions were quenched at different time points by the addition of trifluoroacetate (TFA) to 0.5%. The quenched reaction mixture was diluted into CHCA (α-cyano-4-hydroxycinnamic acid) matrix and spotted on the sample plate. The molecular weight was measured by MALDI-TOF/TOF(SCIEX 5800). All experiments were performed in triplicate.

To measure the kinetic parameters of debenzoylase and deacetylase activity of Hst2, we performed the same MALDI-TOF-based assays. In the debenzoylase activity assay, we used varied H3K9bz peptide concentrations spanning 0.5–10 μM at a fixed 0.1 μM Hst2 enzyme and 1 mM $NAD^+$. In the deacetylase activity assay, we used varied H3K9ac peptide concentrations spanning 2-40 μM at a fixed 0.05 μM Hst2 enzyme and 1 mM $NAD^+$. The background rates in the absence of enzyme were subtracted from the initial velocities derived from the Hst2-catalyzed reactions. Each measurement was performed in triplicate, and only the linear portion of the kinetic traces was used to determine the initial rates. The data were fitted to Michaelis-Menten in GraphPad Prism 8 (GraphPad Software, USA).

**Isothermal titration calorimetry (ITC)**. ITC titrations were performed using a MicroCal ITC200 system (GE Healthcare, USA) at 20 °C. For ITC measurement, peptides and proteins were dialyzed into the same buffer (25 mM Tris-HCl, pH 8.0, 150 mM NaCl). For Hst2-related ITC experiments, peptides at 1.0 mM in the syringe were titrated into Hst2 at 0.1 mM in the sample cell. For reader-related ITC experiments, peptides at 2.0 mM in the syringe were titrated into Taf14[YEATS]/Sas5[YEATS]/Sth1[BD] at 0.1–0.2 mM in the sample cell. Each titration consisted of 20 successive injections. The titration curves were processed using the Origin 7.0 software program (OriginLab) according to the one-binding-site fitting model. Each titration was repeated twice, and one representative plot was shown in the paper. The dissociation constants ($K_d$), enthalpy changes (ΔH), and fitting errors were derived from one representative ITC plot.

**Crystallization and Structural Determination**. Hst2[8-294]-H3K9bz (QTARKbzSTGGK), Taf14[2-137]- H3K9bz (Ac-QTARKbzSTGG), Sas5[2-139]-H3K27bz(KQLASKAARKbzSAPSTGGVKY), and Sth1[1248-1359]-H3K14bz (TARKSTGGKbzAPRKQLASY) complexes were prepared by mixing proteins (15 mg/ml) and peptides in a 1:5 molar ratio. Crystals were obtained by the sitting-drop vapor diffusion method at 25 °C by mixing 300 nL protein-peptide solution and 300 nL crystal solution. The Hst2[8-294]-H3K9bz complex was crystallized in a solution composed of 25% (w/v) PEG 1,500, 0.1 M MMT/sodium hydroxide (pH 6.5). The Hst2[8-294]-H3K9bz-NAD$^+$ complex was crystallized in a solution composed of 0.2 M ammonium fluoride, 20% w/v PEG 3,350. Crystals of Taf14[2-137]-H3K9bz complex were obtained in 48% (v/v) PEG 600 and 0.04 M citric acid (pH 6.0). The Sas5[2-139]-H3K27bz complex was crystallized in 30% (w/v) PEG 4,000, 0.1 M Tris-HCl,pH 8.5, 0.2 M lithium sulfate. The Sth1[1248-1359]-H3K14bz complex was crystallized in a solution containing 25% (w/v) PEG 1,500, 0.1 M MIB/hydrochloric acid,pH 5.0. All crystals were cryoprotected in the same reservoir solution supplemented with 20% glycerol. The datasets were collected at the beamline BL19U1/BL18U1 of the Shanghai Synchrotron Radiation Facility and processed using HKL3000 or HKL2000[49]. The structures were solved by molecular replacement in PHASER[50] by using the following template models: Hst2, PDB 1Q1A; Taf14, PDB 5D7E; Sth1, PDB 6KMJ[26,32,36]. The MR models were then refined by using PHENIX[51] suite with manual building in COOT[52], and the structural details were analyzed by PyMol (1.5.0).

**Bioinformatics analyses**. Subcellular distribution analysis was performed by Uniport (https://www.uniprot.org/). Gene Ontology (GO) analysis was performed by DAVID platform 6.8 (https://david.ncifcrf.gov/). GO categories were selected according to the $P$-values ($P < 0.05$). $P$ values are derived from one-sided Fisher's exact test.

**Statistics and reproducibility**. The experiments of SDS-PAGE and western blot were performed at least twice and similar results were obtained, thus the representative results were shown in Figs. 1a–c, 2a–c, 3a, Supplementary Fig. 3a–d and 7a. The identification of histone and non-histone Kbz sites by LC-MS were performed at least twice and all the repeats are shown in Supplementary data 1–3. The MS/MS identification of histone Kbz sites catalyzed by non-enzymatic or Gcn5-Ada2 subcomplex assays shown in Fig. 2d, e, and the benzoylation rate assays shown in Supplementary Fig. 3f were not repeated due to the material limitation. All other acylation and deacylation assays were repeated for three times, and the

data are presented as mean ± SD ($n = 3$) by using GraphPad Prism 8 software. All ITC experiments were repeated twice, and one representative plot was shown.

**Reporting summary**. Further information on research design is available in the Nature Research Reporting Summary linked to this article.

## Data availability

Coordinates and structure factors reported in the current study have been deposited in the Protein Data Bank under accession codes 7F3S (Sth1$_{BD}$-H3K14bz complex), 7F4A (Taf14$_{YEATS}$–H3K9bz complex), 7F5M (Sas5$_{YEATS}$–H3K27bz complex), 7F4E (Hst2-H3K9bz complex), and 7F5I (Hst2-2′-O-Benzoyl-ADP-ribose complex). The following structures were used in the paper for structural analyses: 5TLR, 6CW2, 1Q1A, 6ISO, 6LS6, 6LSD, 6KMJ, 5D7E, 5IOK, and 6MIQ. The mass spectrometry data have been deposited to the ProteomeXchange Consortium via the PRIDE partner repository with the dataset identifier PXD030110 (LC-MS/MS identification of histones Kbz sites catalyzed by non-enzymatic mechanism and by Gcn5-Ada2), PXD030070 (LC-MS/MS identification of Kbz sites in yeast), and PXD029997 (Global profiling of lysine benzoylation in S. cerevisiae). Source data are provided with this paper. All other data are available from the corresponding author (Yong Chen). Source data are provided with this paper.

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

## Acknowledgements

The authors thank staff members from the BL19U1 and BL18U1 beamlines of the National Facility for Protein Science Shanghai (NFPS) at Shanghai Synchrotron Radiation Facility (SSRF) for X-ray diffraction data collection. We thank the staff members of the Large-scale Protein Preparation System, and the Mass Spectrometry System at the National Facility for Protein Science Shanghai (NFPS) for providing technical support. We thank Jinqiu Zhou for the yeast strains and vectors, Junbiao Dai for the *esa1-531* strain, and Ning Xu for uploading the MS data. This work was supported by grants from the Strategic Priority Research Program of the Chinese Academy of Sciences (XDB37010303 to Y.C.), the National Natural Science Foundation of China (31670748 to Y.C, 31970576 to Y.C., and 31771432 to Y.G.).

## Author contributions

Y.C. conceived and supervised the project. D.W. and F.Y. purified the proteins, carried out crystallization and structural determination, and performed biochemical assays. K.G., M.L., T.L., and Y.G. helped in protein purification and biochemical assays. D.W., P.W., and C.P. performed mass spectrometry analyses. D.W., P.W., C.P., and Y.C. prepared the figures and wrote the manuscript.

## Competing interests

The authors declare no competing interests.
