## [Peer Review File · Nature Communications]

REVIEWER COMMENTS

Reviewer #1 (Remarks to the Author):

Wang et al. carry out a comprehensive study of readers, writers and erasers of lysine benzoylation (Kbz) in yeast, a modification that has been observed in yeast and mammalian cells. In this study proteomic, biochemical and X-ray crystallographic approaches are employed. The authors used proteomic approaches to identify new Kbz histone sites and non-histone Kbz sites in about ~150 proteins. The authors show that Gcn5-containing acetyltransferase complexes and class III NAD-dependent histone deacetylase, including Hst2, could function in vitro as writers and erasers of histone Kbz, respectively. Modeling of Gcn5 is consistent with Kbz binding and crystal structures of Hst2 complexes reveal the molecular basis for Kbz recognition and catalysis by Hst2. The authors also demonstrate that a subset of YEATS domains and Bromodomains can serve as Kbz readers, and crystallographic analyses reveals how YEATS and Bromodomain recognize Kbz marks. Notably, the finding that a bromodomain can recognize the Kbz mark is unexpected given that other studies have not been able to identify such an interaction. Together, these studies implicate several proteins in reading, writing and erasing the Kbz mark, reveal the biochemical and molecular basis for how this can happen, and provide a framework for dissecting the biological functions of the Kbz mark.

Together, this study presents an impressive amount of work identifying new Kbz marks and characterizing the biochemical and structural basis of how the mark is read, written and erased. The studies are rigorous, well described and illustrated, and provides new and important insights into the sites of Kbz modification and the biochemical and structural basis for how proteins could accommodate reading, writing erasing this modification. A significant weakness of this study is that it does not rigorously address the biological and functional importance of the Kbz-related activities that are characterized. Nonetheless, this comprehensive and excellent study is an important contribution to the field and sets the stage for further biological and functional exploration.

The following points should be addressed:

1. The manuscript contains several grammatical errors so should be carefully edited throughout.

2. Line 33; Suggest changing “Our studies reveal the critical regulatory elements for the Kbz pathway...” to “Our studies implicate regulatory elements for the Kbz..” since the biological importance of the Kbz readers, writes and erasers characterized here have not been demonstrated in this study.

3. Line 113; based on the data in Figure 1, the authors state that histone benzoylation is “dynamically regulated” but they do not show any data to support dynamic regulation. The data presented just demonstrates the presence of the modification and does not address the dynamics.

4. Line 125-128; Based on the data in Supplemental Figure 3A, the authors claim that Esa1 does not harbor Kbz activity. Although this appears true against H3 it does not appear true against H2A/H2B so the authors should qualify their statement appropriately.

5. Line 167; The authors claim that removal of Rpd3 has no or negligible effect on histone benzoylation. However, Figure 3a clearly shows that it significantly decreases histone benzoylation. Can the authors rationalize this?

6. Figure 1A; since the pan anti-Kbz is not a common antibody, the authors should include a control in the supplement to show that the antibody is specific for the Kbz mark.

7. Figure 2b, 2c; the relative activity of Gcn5 for acetylation vs. benzoylation is not clear from the Western blots shown so the biological relevance is difficult to assess. The authors should present more quantitative data as they do in Fig. 4d,e vs. Supp Fig 4g,h.

8. Figure 2g; AN additional ITC of succinyl-CoA would nicely complement the data in 2f.

9. Figure 3a; the increase in histone benzoylation due to deletion of class III HDACs (relative to BY4272 or Hos2/3) based on the western blots shown is not as obvious as the authors claim. Can the authors be more quantitative?

10. Figure 3b; the debenzoylase activity assay shown in figure 3b is not very quantitative so it is difficult to assess the biological relevance of the Hst2 debenzoylase activity detected. Can the authors use a more quantitative assay to compare the deacetylase and debenzoylase activities of Hst2.

11. Figure. 3d is not referenced in the text.

12. Figure 4d; It is unclear how the K_d values for H135A, F184A and D68/70A were obtained from the ITC curves shown. H135A and F184A are too flat and D188/70A is not complete and has an unrealistic stoichiometry.

13. Line 253, the authors state that “Mutations of these hydrophobic residues to alanine, including H135, V182, F184, substantially reduced the interactions with both the H3K9bz-containing peptide” but based on the data in Figure 4d, the interaction with V182A is near wild-type. The text needs to be revised to accommodate this finding.

14. Lines 292-293; Figure 5a-c needs to be referenced.

15. Line 473; the authors state that Gcn5-Ada2 complex has “robust benzoyl-CoA binding ability” but I would recommend removing the word “robust” because the data presented (western blots) do not allow the authors to quantitate the levels of lysine benzoylation vs. acetylation.

16. Line 483; suggest changing “long acylation reactions could only be guaranteed in the context of complex formation” to “long acylation reactions are facilitated by other subunits of the Gcn5 complex.”

17. Line 535; Suggest change “indicating the important roles of Kbz in these cellular processes” to “implicating roles of Kbz in these cellular processes” since the biological importance of these modifications by the indicated proteins have not yet been demonstrated.

18. Figure; annotation needs to be in larger font throughout, particularly for the subscripted “Ac” and “Bz,” which are difficult to see, even at 150%.

Reviewer #2 (Remarks to the Author):

The study by Wang et. al. characterizes lysine benzoylation in *S. cerevisiae*, a recently discovered post translational modification in mammalian cells. The authors use biochemical and genetic methods to identify the “readers”, “writers” and “erasers” of lysine benzoylation. While the

identification of readers, writers and erasers for a new PTM would be novel and certainly point towards an in vivo function, I do not think that the data supports the authors conclusions. This study lacks critical controls and the proteomics results do not match the methods used.

Major comments

In vitro benzoylation assay

The authors performed an in vitro assay to determine if Gcn5 complexes could catalyze lysine benzoylation and acetylation (line 129). The authors performed the reaction with enzyme, nucleosome octamer and the acyl-CoA (acetyl or benzoyl). The authors did not show several key control reactions which include a 1. no enzyme control 2. No acyl-CoA control 3. No octamer control.

The no enzyme control is a critical control experiment since acyl-CoAs are known to non-enzymatically modify lysine residues. (refs: <https://doi.org/10.1016/j.celrep.2015.12.030>, <https://doi.org/10.1074/jbc.M113.486753>). The spontaneous reaction of acetyl-CoA and benzoyl-CoA with lysine residues (non-enzymatic reaction) is even more likely given the method for performing the acetylation and benzoylation reactions. In line 657, the authors pre-incubate 2uM of nucleosome octamer with 250 uM of acyl-CoA (acetyl/benzoyl) for 5 min at 30 deg C. These reaction conditions favor spontaneous acylation. Therefore, the data can also be explained by a nonenzymatic mechanism.

The authors performed binding assays to measure the dissociation constant of Gcn5-Ada2 with acetyl- and benzoyl-CoA. The KD is a thermodynamic parameter that does not reflect enzyme catalysis. Additionally, the binding interaction could be driven by the larger CoA moiety of benzoyl-CoA. To show that Gcn5-Ada2 does in fact catalyze lysine benzoylation, the authors should measure kinetic parameters such as Km and Kcat. Another important question to address is the concentration of benzoyl-CoA in cells. Is the concentration of benzoyl-CoA high enough for its utilization as a cosubstrate by Gcn5?

Non-histone lysine benzoylation mass spectrometry

The authors performed a proteomic screen to identify lysine benzoylation on non-histone proteins. They used a common approach for identifying PTMs which includes trypsin digestion followed by immunoaffinity enrichment and mass spectrometry analysis. This approach identified 207 lysine benzoylation sites that were above the 1% FDR threshold.

The database search was performed using PEAKS Studio 8.5 using the variable modifications: 1. Methionine oxidation. 2. Protein N-terminal acetylation. 3. Lysine acetylation. 4. Lysine

monomethylation 5. Lysine dimethylation. 6. Lysine trimethylation. 7. Arginine monomethylation 8. Arginine dimethylation 9. Lysine benzoylation. Additionally, histone samples included the above variable modification along with 10. Lysine propionylation.

Including a high number of variable modifications is problematic because the search space increases exponentially with increasing variable modifications which leads to high false discovery rate (<https://doi.org/10.1021/acs.jproteome.5b00599>). The authors do not provide an unfiltered supplementary table with the peptide spectrum match (PSM) that would include peptide sequence, peptide modified sequence, q-value, pep. These values are critical to assess the quality of the mass spectrum to peptide identification.

I have concerns about some of the ms/ms spectra provided in supplemental regarding the histone benzoylation. For histone analysis, the authors performed two rounds of propionylation. This would modify all unmodified and mono-methylated lysines with a propionyl group. However, some of the ms/ms spectra show unmodified lysines as well as C-terminal lysines (a modified lysine also prevents trypsin cleavage and thus all histone peptides should have a C-terminal Arginine). The peptides in question include H3K9bz, H3K27bz, H3K18bz, H3K23bz, H3K36bz, H3K37bz, H3K115bz, H3K12bz, H4K5bz, H4K8bz, H4K12bz, H2AK4bz, H2AK7bz, H2AZK3bz, H2AZK8bz, H2AZK10bz, H2BK6bz, H2BK11bz, H2BK16bz, H2BK22bz, H2BK37bz, H2BK89bz. In fact, the majority of histone peptides shown have unmodified lysine residues which should otherwise be propionylated. One possible explanation is that these are false hits due to the highly inflated database search space from the many variable modifications.

Additionally, the ms/ms spectra provided do not “look” like high quality spectra. In fact, some ms/ms spectra appear to be electronic noise (Fig 7B, Supp Fig 1 (H4K31bz, H4K12bz, H3K115bz, H3K121bz), Supp Fig 2 (H2BK6bz), Supp Fig 7D).

This study aimed at characterizing the biochemical nature of lysine benzoylation in *S. cerevisiae*. While not the scope of the study, there was no attempt at identifying a functional role for this PTM. The authors identified Hst2 mutants that displayed altered binding and enzyme activity of acetylation and benzoylation. The authors could engineer a cell line where these mutants are expressed to try to dissect the role of debenzoylation vs deacetylation in vivo. A functional characterization of this new modification in yeast would elevate the impact of the study to the level of Nature Communications.

Minor Comments

On line 149, the authors write:

The benzoyl group, which has a similar dimension as the succinyl group, perfectly fits into the catalytic pocket of Gcn5 and makes extensive hydrophobic contacts with Gcn5

Yet, in line 143, they state:

Benzoyl-CoA was built in an identical conformation seen in the succinyl-CoA-hGcn5 complex, except for replacing the terminal succinyl group with a benzoyl group.

Therefore, the modeling results would be expected to fit “perfectly” since this is how the model was made. In the absence of a crystal structure of Gcn5-Ada2 with benzoyl-CoA, the authors should rephrase their strong conclusion statement.

In line 165, the authors write:

In sharp contrast, only a subset of HDAC-deletion strains displayed a considerable increase in global histone benzoylation. Removal of Class I HDAC (Rpd3) or Class II HDAC (Hda1, Hos1, Hos2, Hos3) enzymes had no or negligible effect on Kbz signals. Members of Class III HDACs exhibited different debenzoylation abilities: Kbz levels were increased in *hst2Δ* and *hst3Δ* strains, while no Kbz enhancement was observed in the *sir2Δ* strains (Fig. 3a).

The western blot (Figure 3A) does not show a “considerable” increase in histone benzoylation upon deletion of different HDACs. In fact, no quantification is done to determine the degree of change.

The authors performed a time course for debenzoylation and deacetylation with Hst2 (Figure 3B and 3C). The enzyme concentrations between the two reactions (deacetyl and debenzoyl) are not the same (2uM for debenzoylation vs 0.5uM for deacetylation). The authors should show the kinetics between deacetylation and debenzoylation using equal enzyme concentration to directly compare kinetic rates. Additionally, the authors measured dissociation constants for Hst2 with different benzoylated peptides (Figure 3F). The authors should also show K_m and K_{cat} values for Hst2 and the acetyl and benzoyl peptide substrates for comparison.

Reviewer #3 (Remarks to the Author):

In the manuscript entitled ‘Global profiling of regulatory elements in the histone benzoylation pathway’ Wang et al., investigated the presence of lysine benzoylation (Kbz) in yeast. They further tried to uncover the writers, erasers, and readers of this mark. Furthermore, they obtained crystal structures that reveal the molecular basis of the engagement between the eraser and reader with

the Kbz mark. The work is definitely of interest to the field. Overall, the experiments are well-done and the quality of the data is good. Also, the manuscript flows well.

However, certain things are holding back the manuscript. This includes a lack of experimental details, controls, and appropriate discussion. In multiple instances, conclusions are drawn without providing direct experimental evidence, some of which should be very easy for the authors to do. The biggest concern is the lack of supportive experiments performed in yeast to substantiate their in vitro experiments.

This reviewer feels that this manuscript will be a strong candidate for publication in Nature Communications once the following concerns have been addressed:

1. The results should be more descriptive. What is XI octamer in Figure 1a? Was E.coli used as a negative control? Some of these things are mentioned in the legends section but should also be included in the main text.
2. Figure 1B: Experimental details are missing. For how long the cells were treated with sodium benzoate? Was BY4741 used? This information must be added in figure legends at least.
3. Line 90-92: Especially considering that the antibody used in pan Kbz, the data is not clear enough to make such a conclusion. Speculation is ok. But the basis of the speculation should be described.
4. Figure 1C: Again, very poorly described experiment. What does H₂O mean? Cells were moved from YPD to water? Yeast won't grow in water and will be stagnant. So does that mean that stagnant yeast loses the mark?
5. Line 102: For how long were the cells treated with sodium benzoate?
6. Line 105: What happens to H3K9bz, H3K14bz, H3K27bz, and H4K44bz when yeast are grown in 4% EtOH?
7. Mass spec data: The results need to be more descriptive. How much percentage of peptides were modified? How many replicates?
8. Line 167-171: The authors state that 'Removal of Class I HDAC (Rpd3) or Class II HDAC (Hda1, Hos1, Hos2, Hos3) enzymes had no or negligible effect on Kbz signals.' I do not agree with this conclusion as the western blot data in Figure 3a doesn't support this. Rpd3 Δ almost completely abolishes Kbz. Also, the changes in Kbz signal appear similar in Hos1 and Hos2 deletion as that seen upon hst2 and 3 deletions.
9. Figure 3b-f: Using peptides in in vitro assays the authors profile the substrate specificity of Hst2-mediated debenzoylation. This data should be completed with mass spec analysis of histones from the Hst2 Δ BY4742 strain as well. In fact, many of the in vitro assays presented in this manuscript to test activity will be benefited by complementing them with experiments done in yeast cells.
10. Line 186-188: Citing the data shown in figure 1, the authors conclude that H3 is the most extensively benzoylated histone among core histones. However, based on figure 1 this cannot be

concluded. What percentage of peptides were modified? The results from western blot might be due to the lower affinity of the antibody for H4.

11. Line 199-201: The authors state that 'critical arginine residue adjacent to the target lysine site determining the tight binding between Hst2 and histone substrates'. The authors should mutate R and do ITC to provide direct evidence for this. Also, what is the importance of R structurally? Should be discussed.

12. Line 306-309: The authors state that 'Sas5YEATS may be a favorable Kbz reader in yeast cells'. Again, no in vivo data to support this.

13. Line 416-419: Again, the important information of how much percentage of peptides were modified is missing.

14. Line 438-440: In the absence of any direct evidence, this statement is just speculation and should not be phrased as a conclusion.

15. Figure 2a: Why there is an increase in Kbz in the Hpa1 deletion strain? Should be discussed.

16. Figure 2b: Why band shift in anti-H3N samples?

17. Figure 2b and 2c: There is no control that lacks Ada2 Gcn5 as well as SAGA.

18. Figure 2b and 2c: Which histone sites are being modified by Ada2 Gcn5 and SAGA? Mass spec data will be useful.

19. Figure 3a: Many of the differences are subtle. The experiment should be repeated and quantification plots should be added. Also, the H4 signal appears to be saturated. A lower exposure blot would be appreciated.

20. Figure 3a: Why does sir2 deletion result in a decrease in Kbz? Discussion or speculation would be welcome.

21. Figure 3e: Unmodified H3 peptide should be included as control in the ITC experiment.

22. Figure 5a and b: Based on the crystal structure, please explain why the binding specificities might be different.

23. Figure 5C: Again, no evidence was provided from experiments done in yeast.

24. Does treatment of yeast cells with sodium benzoate alter global H3 acetylation level?

25. The structural work by the authors reveals that the mode of binding to acetyl and benzoyl groups is very similar. Do the acetyl and benzoyl groups compete for binding?

Minor comments:

1. The manuscript will be benefitted from thorough proofreading.

a. Line 41: Was initially identified, not is.

- b. Line 56: is unclear, not are.
 - c. Line 62: is, not are.
 - d. Line 64: in, not from.
 - e. Line 77: performed, not perform.
 - f. Line 944: series, not serial
 - g. Line 257: examined, not examine
 - h. Line 267: ensures, not assures
2. Mention Kac stands for lysine acetylation.
 3. Line 52: I think at this stage of the manuscript, it is better to replace histone acetyltransferase with enzymes. Because otherwise, it shows an inherent bias that HATs will be responsible for histone benzoylation.
 4. Line 68-71: Why suddenly talk about Khib? I feel that it is distracting.
 5. There should be a separate section for antibodies under Material and Methods.
 6. Figures overall, especially the font size should be bigger for easy readability.
 7. Line 162: Use of the phrase 'deletion mutants' is more common in yeast than 'knock-out'.
 8. Line 420-424: How was this determined? Which software or server was used?
 9. Figure 4f should come after 4C. However, this is an optional change.
 10. Line 173: Purified from where?
 11. Line 289-290: This sentence should be phrased better.

Response to Reviewer #1

Wang et al. carry out a comprehensive study of readers, writers and erasers of lysine benzoylation (Kbz) in yeast, a modification that has been observed in yeast and mammalian cells. In this study proteomic, biochemical and X-ray crystallographic approaches are employed. The authors used proteomic approaches to identify new Kbz histone sites and non-histone Kbz sites in about ~150 proteins. The authors show that Gcn5-containing acetyltransferase complexes and class III NAD-dependent histone deacetylase, including Hst2, could function in vitro as writers and erasers of histone Kbz, respectively. Modeling of Gcn5 is consistent with Kbz binding and crystal structures of Hst2 complexes reveal the molecular basis for Kbz recognition and catalysis by Hst2. The authors also demonstrate that a subset of YEATS domains and Bromodomains can serve as Kbz readers, and crystallographic analyses reveals how YEATS and Bromodomain recognize Kbz marks. Notably, the finding that a bromodomain can recognize the Kbz mark is unexpected given that other studies have not been able to identify such an interaction. Together, these studies implicate several proteins in reading, writing and erasing the Kbz mark, reveal the biochemical and molecular basis for how this can happen, and provide a framework for dissecting the biological functions of the Kbz mark.

Together, this study presents an impressive amount of work identifying new Kbz marks and characterizing the biochemical and structural basis of how the mark is read, written and erased. The studies are rigorous, well described and illustrated, and provides new and important insights into the sites of Kbz modification and the biochemical and structural basis for how proteins could accommodate reading, writing erasing this modification. A significant weakness of this study is that it does not rigorously address the biological and functional importance of the Kbz-related activities that are characterized. Nonetheless, this comprehensive and excellent study is an important contribution to the field and sets the stage for further biological and functional exploration.

We thank this reviewer for the appreciation of our work and constructive suggestions.

1. The manuscript contains several grammatical errors so should be carefully edited throughout.

Thank you for this suggestion. The manuscript has been professionally edited by a native speaker.

2. Line 33; Suggest changing "Our studies reveal the critical regulatory elements for the Kbz pathway..." to "Our studies implicate regulatory elements for the Kbz.." since the biological importance of the Kbz readers, writes and erasers characterized here have not been demonstrated in this study.

Thank you for this suggestion. In the revision, we have updated this sentence as

follows: "Our studies implicate regulatory elements for Kbz and provide a framework for dissecting the biological functions of lysine benzoylation."

3. Line 113; based on the data in Figure 1, the authors state that histone benzoylation is "dynamically regulated" but they do not show any data to support dynamic regulation. The data presented just demonstrates the presence of the modification and does not address the dynamics.

We are sorry for the confusion. As shown in Fig. 1c, we initially wanted to reveal that "the level of histone Kbz modification is significantly different in the presence of four carbon sources, indicating that histone Kbz is dynamically regulated by cellular metabolism." We agree with your comments and have revised this sentence to precisely reflect what we observed: "Collectively, these results suggest that histone benzoylation is an evolutionarily conserved marker and that the benzoylation level in yeast is sensitive to carbohydrate metabolism."

4. Line 125-128; Based on the data in Supplemental Figure 3A, the authors claim that Esa1 does not harbor Kbz activity. Although this appears true against H3 it does not appear true against H2A/H2B so the authors should qualify their statement appropriately.

Thank you for this suggestion. In the revision, we have revised the description of Esa1 data as follows: "The acetylation levels of all histones were completely abolished in the *esa1-531* mutant at the non-permissive temperature (37 °C). In sharp contrast, the Kbz level of H3 was only slightly reduced, and the Kbz levels of H2A/H2B were severely decreased at the non-permissive temperature, indicating that Esa1 might have weak benzoyltransferase activity specific to H2A/H2B (Supplementary Fig. 3a)".

5. Line 167; The authors claim that removal of Rpd3 has no or negligible effect on histone benzoylation. However, Figure 3a clearly shows that it significantly decreases histone benzoylation. Can the authors rationalize this?

We thank this reviewer for raising this point. We were also surprised that Rdp3 deletion led to a significant decrease in histone benzoylation. We have revised the description of this phenomenon as follows: "Unexpectedly, the removal of some HDACs, including Rpd3, Hos3, and Sir2, decreased the global histone benzoylation levels (Fig. 3a). These enzymes might be involved in the benzoyl-CoA metabolic pathway or regulate the expression levels of histone benzoyltransferases, which merits further investigation".

6. Figure 1A; since the pan anti-Kbz is not a common antibody, the authors should include a control in the supplement to show that the antibody is specific for the Kbz mark.

We thank the reviewer for the suggestion and have included more controls in the revision. We have included the H3K9cr and H3K9ac peptides to show that the Kbz antibody does not recognize other acylated peptides (Supplementary Fig. 1a). Moreover, as shown in Fig. 1a, the pan anti-Kbz antibody does not recognize recombinant histones purified from *E.coli*, further confirming the specificity of this antibody.

7. Figure 2b, 2c; the relative activity of Gcn5 for acetylation vs. benzoylation is not clear from the Western blots shown so the biological relevance is difficult to assess. The authors should present more quantitative data as they do in Fig. 4d,e vs. Supp Fig 4g,h.

We thank the reviewer's suggestion. In the past three months, we tried many different assay systems to compare the activity difference between acetyltransferase activity and benzoyltransferase activity of Gcn5-Ada2. However, the benzoylation activity of Gcn5-Ada2 was much weaker than the acetylation activity (Supplementary Fig. 3e-h). We also list these data here as Reference Fig. 1 for easy access by reviewers. The rough estimation is that the benzoylation rate under our assay conditions was only 1/520 of the acetylation rate, so we could not obtain reliable signals to measure the kinetic parameters of the Gcn5-catalyzed benzoylation reaction. We have clearly stated this point in the revised manuscript in lines 158-160.

Reference Figure 1. The acylation activity assay was performed by measuring the generation of HS-CoA by DNTB. The protein (Gcn5-Ada2) concentration is $2 \mu\text{M}$. The cofactor acetyl-CoA or benzoyl-CoA is $500 \mu\text{M}$, and the substrate (H3₁₋₁₈ peptide) is $250 \mu\text{M}$.

- Comparison of the acetylation and benzoylation activity of Gcn5-Ada2. Gcn5-Ada2 showed strong acetylation activity but no detectable benzoylation activity in the 3-min time frame. The acetylation rate at the current reaction condition was $10.35 \mu\text{M}/\text{min}$.
- Gcn5-Ada2 only showed detectable benzoylation activity after a long-time reaction (more than 1 hour). At the current reaction condition, the benzoylation rate was $1.19 \mu\text{M}/\text{h}$, which was only 1/520 of the acetylation rate.

8. Figure 2g; An additional ITC of succinyl-CoA would nicely complement the data in 2f.

Thanks for this suggestion. In the revision, we performed the ITC analysis for Gcn5-Ada2 binding succinyl-CoA, and the determined K_D was 14.5 μ M, which was weaker than that for benzoyl-CoA. The data are comparatively presented in the new Supplementary Fig. 3j.

9. Figure 3a; the increase in histone benzoylation due to deletion of class III HDACs (relative to BY4272 or Hos2/3) based on the western blots shown is not as obvious as the authors claim. Can the authors be more quantitative?

We thank the suggestion of WB quantification. In the revision, we have re-performed the western blot assays with the widely used Odyssey Infrared Imaging system, giving accurate quantitative western blot results with near-infrared fluorescence. Near-infrared (NIR) fluorescence delivers consistent signals not affected by exposure timing. Figs. 2a and 3a, which screened the benzoyltransferases and debenzoylase, were updated. In the new Fig. 2a and 3a, the fluorescence signals from mutant strains were normalized by setting the WT fluorescence signal to 1.0 and were labeled in the image. According to the new quantification data, we have rephrased our description about HDAC-deletion data as follows: "In sharp contrast, deletion of HDACs only mildly increased the benzoylation levels less than 2-fold (Fig. 3a). Removal of Hst2 caused the highest increase of the H3 Kbz signal around 1.8-fold (Fig. 3a). Unexpectedly, the removal of some HDACs, including Rpd3, Hos3, and Sir2, decreased the global histone benzoylation levels (Fig. 3a). These enzymes might be involved in the benzoyl-CoA metabolic pathway or regulate expression levels of histone benzoyltransferases, which merits further investigation."

10. Figure 3b; the debenzoylase activity assay shown in figure 3b is not very quantitative so it is difficult to assess the biological relevance of the Hst2 debenzoylase activity detected. Can the authors use a more quantitative assay to compare the deacetylase and debenzoylase activities of Hst2.

We thank the reviewer for raising this issue. We performed kinetic experiments using H3K9ac and H3K9bz peptides, respectively. The results presented in Figs. 3d and 3e showed that the K_m for deacetylation was approximately 6-fold higher than that for debenzoylation, and the turnover number (k_{cat}) for deacetylation was 63.53-fold higher than that for debenzoylation. Therefore, k_{cat}/K_m for deacetylation was approximately 10.6-fold for debenzoylation. Hst2 was more effective in deacetylation than in debenzoylation in our assay system.

11. Figure. 3d is not referenced in the text.

We are sorry for this mistake. The original Fig. 3d, which is Fig. 3f in the revised figure, has been correctly referenced in the main text as follows: "Hst2 could remove the benzoylation markers from all the peptides tested, albeit with different efficiencies (Fig. 3f)."

12. Figure 4d; It is unclear how the K_D values for H135A, F184A and D68/70A were obtained from the ITC curves shown. H135A and F184A are too flat and D188/70A is not complete and has an unrealistic stoichiometry.

We thank the reviewer for raising this issue. Although the ITC curves for H135A and F184A looked flat when merged with the WT curve, their signals (ΔH) were strong enough to obtain a relatively good fit. The original ITC titration figures for H135A and F184A mutations are shown below for the reviewer's reference (Reference Fig. 2a, 2b). Moreover, for Hst2 D68/D70A mutant, we have adjusted the protein concentration and performed ITC again to reach saturated binding (Reference Fig. 2c). The new K_D value for Hst2^{D68/D70A} binding H3K9bz is 17.89 μM , similar to what we measured before. Accordingly, Fig. 4e is updated in the revised manuscript.

Reference Figure 2. ITC analyses of Hst2 with H3K9bz peptides.

- ITC analysis of Hst2 H135A mutant binding with H3K9bz.
- ITC analysis of Hst2 F184A mutant binding with H3K9bz.
- ITC analysis of Hst2 D68A/D70A mutant binding with H3K9bz.

13. Line 253, the authors state that "Mutations of these hydrophobic residues to alanine, including H135, V182, F184, substantially reduced the interactions with both the H3K9bz-containing peptide" but based on the data in Figure 4d, the interaction with V182A is near wild-type. The text needs to be revised to accommodate this finding.

Thanks for this suggestion. In the revision, we have revised the text to accommodate the findings as follows: "Mutations of these hydrophobic residues to alanine, including H135, V182, and F184, reduced the interactions with both the

H3K9bz-containing peptide and the H3K9ac-containing peptide to varying degrees: H135A and F184A severely disrupted the interaction with acylated peptides, while V182 only mildly decreased the binding approximately 3-fold."

14. Lines 292-293; Figure 5a-c needs to be referenced.

We are sorry for this mistake. In the revised manuscript, Figs. 5a-c have been referenced as follows: "ITC analyses showed that both Taf14 and Sas5 have Kbz binding abilities, albeit with different specificities. Sas5^{YEATS} prefers binding with H3K14 and H3K27, whereas Taf14 showed comparatively strong binding toward H3K9bz, K18bz, and K27bz (Fig. 5a, 5b). In sharp contrast, Yaf9^{YEATS} exhibited no detectable binding to any tested Kbz peptide (Fig. 5c), although Yaf9^{YEATS} could bind to H3K27ac peptide with a similar K_d (K_d = 280 μM) as reported before²⁶(Supplementary Fig. 5a)."

15. Line 473; the authors state that Gcn5-Ada2 complex has "robust benzoyl-CoA binding ability" but I would recommend removing the word "robust" because the data presented (western blots) do not allow the authors to quantitate the levels of lysine benzoylation vs. acetylation.

Thank you for this suggestion. Indeed, Gcn5-Ada2 had much weaker benzoyltransferase activity than acetyltransferase activity (Supplementary Fig. 3e-f). In the revision, we have modified this sentence as follows: "Gcn5-Ada2 complex has comparable benzoyl-CoA and succinyl-CoA binding affinities." We also clearly stated the fact that "the benzoylation activity of Gcn5-Ada2 is much weaker than the acetylation activity" in the revised manuscript.

16. Line 483; suggest changing "long acylation reactions could only be guaranteed in the context of complex formation" to "long acylation reactions are facilitated by other subunits of the Gcn5 complex."

Thanks for this suggestion. In the revision, we have changed the text according to the reviewer's suggestion as follows: "long acylation reactions are facilitated by other subunits of the Gcn5 complex."

17. Line 535; Suggest change "indicating the important roles of Kbz in these cellular processes" to "implicating roles of Kbz in these cellular processes" since the biological importance of these modifications by the indicated proteins have not yet been demonstrated.

Thank you for this suggestion. In the revision, we have changed this sentence as follows: "implicating potential roles of Kbz in these cellular processes."

18. Figure; annotation needs to be in larger font throughout, particularly for the

subscripted "Ac" and "Bz," which are difficult to see, even at 150%.

We are sorry for the labeling issue. In the revision, we have updated labels in all the figures to ensure that readers can see them at the printed size clearly.

Response to Reviewer #2:

The study by Wang et. al. characterizes lysine benzoylation in *S. cerevisiae*, a recently discovered post translational modification in mammalian cells. The authors use biochemical and genetic methods to identify the "readers", "writers" and "erasers" of lysine benzoylation. While the identification of readers, writers and erasers for a new PTM would be novel and certainly point towards an in vivo function, I do not think that the data supports the authors conclusions. This study lacks critical controls and the proteomics results do not match the methods used.

We thank this reviewer's criticism, correction, and suggestion for our work. We thank you for pointing out the weakness of our manuscript and giving constructive advice to help improve our manuscript. We have performed additional experiments and substantially revised the manuscript in the revision. We hope the current manuscript can satisfactorily address the concerns raised.

Major comments

In vitro benzoylation assay

The authors performed an in vitro assay to determine if Gcn5 complexes could catalyze lysine benzoylation and acetylation (line 129). The authors performed the reaction with enzyme, nucleosome octamer and the acyl-CoA (acetyl or benzoyl). The authors did not show several key control reactions which include a 1. no enzyme control 2. No acyl-CoA control 3. No octamer control. The no enzyme control is a critical control experiment since acyl-CoAs are known to non-enzymatically modify lysine residues.

(refs: <https://doi.org/10.1016/j.celrep.2015.12.030>, <https://doi.org/10.1074/jbc.M113.486753>). The spontaneous reaction of acetyl-CoA and benzoyl-CoA with lysine residues (non-enzymatic reaction) is even more likely given the method for performing the acetylation and benzoylation reactions. In line 657, the authors pre-incubate 2uM of nucleosome octomer with 250 uM of acyl-CoA (acetyl/benzoyl) for 5 min at 30 deg C. These reaction conditions favor spontaneous acylation. Therefore, the data can also be explained by a non-enzymatic mechanism.

We thank this reviewer's insightful suggestion. This reviewer is correct. The non-enzymatic benzoylation indeed exists in our assay system. According to this reviewer's suggestion, we have compared the benzoylation reaction in the presence and absence of Gcn5-Ada2. The result is shown in new Fig. 2c and also shown below as Reference Fig. 3a. In the absence of enzymes, we could still see increased benzoylation signals. Moreover, the LC-MS/MS data confirmed the western blot data that histone benzoylation could spontaneously occur on histones in the absence of enzymes, but Gcn5-Ada2 could induce more benzoylated histone sites and catalyze benzoylation at a higher abundance than that without enzymes (Fig. 2d-e, also shown

as Reference Fig. 3b-c). Therefore, we propose that lysine benzoylation can occur by both non-enzymatic and enzymatic mechanisms.

Reference Figure 3. Comparison of enzyme-dependent and enzyme-independent histone benzoylation.

- Non-enzymatic and enzymatic histone benzoylation reactions are shown by western blot. Both showed gradually increased benzoylation signals.
- The list of LC-MS/MS-identified histone benzoylation sites deposited by the non-enzymatic mechanism and by Gcn5-Ada2. The unique sites catalyzed by Gcn5-Ada2 are labeled red.
- Gcn5-Ada2 catalyzed histone benzoylation more efficiently than spontaneous reaction. The Y-axis represents the ratio of MS/MS peak areas under enzymatic and non-enzymatic conditions.

The authors performed binding assays to measure the dissociation constant of Gcn5-Ada2 with acetyl- and benzoyl-CoA. The KD is a thermodynamic parameter that does not reflect enzyme catalysis. Additionally, the binding interaction could be driven by the larger CoA moiety of benzoyl-CoA. To show that Gcn5-Ada2 does in fact catalyze lysine benzoylation, the authors should measure kinetic parameters such as Km and Kcat. Another important question to address is the concentration of benzoyl-CoA in cells. Is the concentration of benzoyl-CoA high enough for its utilization as a cosubstrate by Gcn5?

We totally agree with the reviewer's comments. The binding affinity with acetyl-CoA or benzoyl-CoA does not reflect enzyme catalytic efficiency. Indeed, the acetylation activity of Gcn5-Ada2 is much higher than benzoylation (see Reference Figure 1 in the reply for reviewer 1). The rough estimation is that the benzoylation rate under our assay conditions was only 1/520 of the acetylation rate. The extremely low benzoylation activity of Gcn5-Ada2 prevents us from measuring the kinetic parameters of Gcn5-catalyzed benzoylation reactions. We have clearly stated this observation in the revised manuscript in lines 158-160.

According to the reviewer's suggestion, we quantified the cellular abundance of benzoyl-CoA in yeast BY4742 strain treated with 10 mM sodium benzoate for 6 h using an LC-MS/MS method. The data showed that the cellular benzoyl-CoA concentration was approximately 0.33 μ M (296.35 ng/ml) (Supplementary Fig. 1b). For comparison, the acetyl-CoA level was about 3-30 μ M (2.43-24.3 μ g/ml) in yeast cells¹. Therefore, the concentration of acetyl-CoA is 10-100 fold greater than that of benzoyl-CoA. In a previous study, Li et al. showed that the cellular concentration of

malonyl-CoA was around 2.5% of that of acetyl-CoA in yeast². One study showed that the cellular propionyl-CoA concentration was 92 ng/ml, and acetyl-CoA was 754 ng/ml in 293T cells, and the abundance of propionyl-CoA was 12 % of that of acetyl-CoA³. Another study showed that the contents of crotonyl-CoA were approximately 1/1000 of acetyl-CoA in HeLa S3 cells⁴. Our data suggested that benzoyl-CoA was in a similar concentration range of other rare metabolites (crotonyl-CoA, malonyl-CoA, or propionyl-CoA). Because the low concentration of crotonyl-CoA can still support histone crotonylation and play important physiological roles in cells, we believe that even if the cellular concentration of benzoyl-CoA is lower than the K_m determined by in vitro assays, benzoylation could still occur in cells.

Non-histone lysine benzoylation mass spectrometry

The authors performed a proteomic screen to identify lysine benzoylation on non-histone proteins. They used a common approach for identifying PTMs which includes trypsin digestion followed by immunoaffinity enrichment and mass spectrometry analysis. This approach identified 207 lysine benzoylation sites that were above the 1% FDR threshold. The database search was performed using PEAKS Studio 8.5 using the variable modifications: 1. Methionine oxidation. 2. Protein N-terminal acetylation. 3. Lysine acetylation. 4. Lysine monomethylation 5. Lysine dimethylation. 6. Lysine trimethylation. 7. Arginine monomethylation 8. Arginine dimethylation 9. Lysine benzoylation. Additionally, histone samples included the above variable modification along with 10. Lysine propionylation. Including a high number of variable modifications is problematic because the search space increases exponentially with increasing variable modifications which leads to high false discovery rate (<https://doi.org/10.1021/acs.jproteome.5b00599>). The authors do not provide an unfiltered supplementary table with the peptide spectrum match (PSM) that would include peptide sequence, peptide modified sequence, q-value, pep. These values are critical to assess the quality of the mass spectrum to peptide identification.

We thank this reviewer's suggestion. In the revision, we have provided the corresponding information required to assess the MS data quality in Supplementary Table 2. For data inspection and assessment, the mass spectrometry data have been deposited to the ProteomeXchange Consortium via the PRIDE partner repository with the dataset identifier PXD030070 (LC-MS/MS identification of histone Kbz sites in *S. cerevisiae*) and PXD029997 (global profiling of lysine benzoylation in *S. cerevisiae*).

I have concerns about some of the ms/ms spectra provided in supplemental regarding the histone benzoylation. For histone analysis, the authors performed two rounds of propionylation. This would modify all unmodified and mono-methylated lysines with a propionyl group. However, some of the ms/ms spectra show unmodified lysines as well as C-terminal lysines (a modified lysine also prevents trypsin cleavage and thus all histone peptides should have a C-terminal Arginine). The peptides in question include H3K9bz, H3K27bz, H3K18bz, H3K23bz, H3K36bz, H3K37bz, H3K115bz,

H3K12bz, H4K5bz, H4K8bz, H4K12bz, H2AK4bz, H2AK7bz, H2AZK3bz, H2AZK8bz, H2AZK10bz, H2BK6bz, H2BK11bz, H2BK16bz, H2BK22bz, H2BK37bz, H2BK89bz. In fact, the majority of histone peptides shown have unmodified lysine residues which should otherwise be propionylated. One possible explanation is that these are false hits due to the highly inflated database search space from the many variable modifications.

We thank this reviewer for raising this issue. It is our fault that we did not provide sufficient details about our sample preparation procedure in the initial submission. To promote the quality and depth of histone benzylation, we performed MS experiments in two different ways, one with two rounds of propionylation and another with reductive alkylation without any propionylation. All the samples were subjected to LC-MS/MS analysis. Then, we manually checked the quality of these identified histone benzylated peptides and selected the MS/MS spectrum with better quality. The results showed that the quality of the spectrum from reductive alkylation was better than that with two rounds of propionylation. So the histone benzylation data presented in this manuscript and the deposited dataset on ProteomeXchange (project accession: PXD030070) are all from the samples prepared by reductive alkylation without propionylation. Therefore, none of the identified histone peptides showed propionylation. To make it clear, we have modified the related method section accordingly (lines 682-688).

Additionally, the ms/ms spectra provided do not "look" like high quality spectra. In fact, some ms/ms spectra appear to be electronic noise (Fig 7B, Supp Fig 1 (H4K31bz, H4K12bz, H3K115bz, H3K121bz), Supp Fig 2 (H2BK6bz), Supp Fig 7D).

Thank you so much for this suggestion. Due to the low abundance of lysine benzylation, the signal of most of the modified peptides is weak. The quality of the corresponding MS and MS/MS spectra is not that "great." Nevertheless, both the MS and MS/MS spectra were acquired by an Orbitrap analyzer with high resolution to improve the confidence of the conclusion. To enhance the quality of our study, we have re-evaluated and manually checked the quality of these spectra, replaced the MS/MS spectra of H4K12, H4K31, H3K115 with better spectra, and excluded modification sites with low-quality MS data, such as H2BK6 and H3K121 (Supplementary Fig. 2). The histone benzylation sites after data sorting are 27 sites.

This study aimed at characterizing the biochemical nature of lysine benzylation in *S. cerevisiae*. While not the scope of the study, there was no attempt at identifying a functional role for this PTM. The authors identified Hst2 mutants that displayed altered binding and enzyme activity of acetylation and benzylation. The authors could engineer a cell line where these mutants are expressed to try to dissect the role of debenzylation vs deacetylation in vivo. A functional characterization of this new modification in yeast would elevate the impact of the study to the level of Nature Communications.

We thank this reviewer's suggestion and completely agree that our Hst2 mutants could serve as potential tools to dissect the roles of histone benzoylation in vivo. We tried to engineer the cell lines in which Hst2^{WT}, Hst2^{I117F}, and Hst2^{I117V} mutants were expressed in the *hst2Δ* strains to examine the phenotypes of yeast strains. The preliminary data showed that the Hst2 mutation of I117F and I117V did not generate any growth defects at various temperatures or under different DNA-damage stress conditions (Reference Fig. 4). More detailed analyses, such as RNA-Seq and ChIP-Seq, might be required to investigate the roles of Hst2 mutants in vivo. However, these experiments need extensive time to be performed and analyzed and might be beyond the scope of this study. Thus, we choose not to include these functional analyses in the current manuscript. We hope this reviewer will understand our choice.

Reference Figure 4. Genetic analyses of the Hst2 mutant strains.

Spotting assays with *hst2Δ* strains transformed with plasmids expressing full-length wild-type Hst2, I117F or I117V mutants grown at different temperatures or on plates containing hydroxyurea (HU), methyl methanesulfonate (MMS), and phleomycin (Phle) at 30 °C.

Minor Comments

On line 149, the authors write:

The benzoyl group, which has a similar dimension as the succinyl group, perfectly fits into the catalytic pocket of Gcn5 and makes extensive hydrophobic contacts with Gcn5

Yet, in line 143, they state: Benzoyl-CoA was built in an identical conformation seen in the succinyl-CoA-hGcn5 complex, except for replacing the terminal succinyl group with a benzoyl group. Therefore, the modeling results would be expected to fit "perfectly" since this is how the model was made. In the absence of a crystal structure of Gcn5-Ada2 with benzoyl-CoA, the authors should rephrase their strong conclusion

statement.

Thanks for this suggestion. We have revised this part as follows: "Benzoyl-CoA was built in a conformation seen in the succinyl-CoA-hGcn5 complex, except for replacing the terminal succinyl group with a benzoyl group. In the modeled structure (Supplementary Fig. 3h), the pyrophosphate group and 3'-phosphate-adenosine moiety of CoA sit in a positively charged pocket, while the benzoyl group and pantetheine moiety are docked to a relatively hydrophobic groove of Gcn5, supporting the hypothesis that the active site of the Gcn5 complex has adequate space to accommodate the large benzoyl group (Supplementary Fig. 3h-3i). Consistent with the structural model, the Gcn5-Ada2 complex has comparable benzoyl-CoA and succinyl-CoA binding affinities (Supplementary Fig. 3j)."

In line 165, the authors write:

In sharp contrast, only a subset of HDAC-deletion strains displayed a considerable increase in global histone benzoylation. Removal of Class I HDAC (Rpd3) or Class II HDAC (Hda1, Hos1, Hos2, Hos3) enzymes had no or negligible effect on Kbz signals. Members of Class III HDACs exhibited different debenzoylation abilities: Kbz levels were increased in *hst2Δ* and *hst3Δ* strains, while no Kbz enhancement was observed in the *sir2Δ* strains (Fig. 3a). The western blot (Figure 3A) does not show a "considerable" increase in histone benzoylation upon deletion of different HDACs. In fact, no quantification is done to determine the degree of change.

Thank you for this suggestion. In the revision, we re-performed the western blot assays with the widely used Odyssey Infrared Imaging system, which gives accurate quantitative western blot results with near-infrared fluorescence. Figs. 2a and 3a, which screened the benzoyltransferases and debenzoylases, were updated. In the new Figs. 2a and 3a, the fluorescence signals from mutant strains were normalized by setting the WT fluorescence signal to 1.0 and were labeled in the image. According to the new quantification data, we have rephrased our description about HDAC-deletion data as follows: "In sharp contrast, deletion of most HDACs only mildly increased the benzoylation level less than 2-fold (Fig. 3a). Removal of Hst2 caused the highest increase in the H3 Kbz signal by approximately 1.8-fold (Fig. 3a). Unexpectedly, the removal of some HDACs, including Rpd3, Hos3, and Sir2, decreased the global histone benzoylation levels (Fig. 3a). These enzymes might be involved in the benzoyl-CoA metabolic pathway or regulate expression levels of histone benzoyltransferases, which merits further investigation."

The authors performed a time course for debenzoylation and deacetylation with Hst2 (Figure 3B and 3C). The enzyme concentrations between the two reactions (deacetyl and debenzoyl) are not the same (2uM for debenzoylation vs 0.5uM for deacetylation). The authors should show the kinetics between deacetylation and debenzoylation using equal enzyme concentration to directly compare kinetic rates. Additionally, the authors measured dissociation constants for Hst2 with different benzoylated peptides

(Figure 3F). The authors should also show K_m and K_{cat} values for Hst2 and the acetyl and benzoyl peptide substrates for comparison.

We thank the reviewer for raising this issue. We performed kinetic experiments using H3K9ac and H3K9bz peptides, respectively. The results presented in Figs. 3d and 3e showed that the K_m for deacetylation was about 6-fold higher than that for debenzoylation, while the turnover number (k_{cat}) for deacetylation was 63.53-fold higher than that for debenzoylation. Therefore, k_{cat}/K_m for deacetylation was approximately 10.6-fold for debenzoylation. Hst2 is more effective in deacetylation than in debenzoylation in our assay system.

Response to Reviewer #3:

In the manuscript entitled 'Global profiling of regulatory elements in the histone benzoylation pathway' Wang et al., investigated the presence of lysine benzoylation (Kbz) in yeast. They further tried to uncover the writers, erasers, and readers of this mark. Furthermore, they obtained crystal structures that reveal the molecular basis of the engagement between the eraser and reader with the Kbz mark. The work is definitely of interest to the field. Overall, the experiments are well-done and the quality of the data is good. Also, the manuscript flows well.

However, certain things are holding back the manuscript. This includes a lack of experimental details, controls, and appropriate discussion. In multiple instances, conclusions are drawn without providing direct experimental evidence, some of which should be very easy for the authors to do. The biggest concern is the lack of supportive experiments performed in yeast to substantiate their in vitro experiments.

This reviewer feels that this manuscript will be a strong candidate for publication in Nature Communications once the following concerns have been addressed:

We thank this reviewer for the appreciation of our work and constructive suggestions.

1. The results should be more descriptive. What is XI octamer in Figure 1a? Was *E.coli* used as a negative control? Some of these things are mentioned in the legends section but should also be included in the main text.

Thank you for this suggestion. XI stands for *Xenopus laevis*. We have revised the main text and figure legend of Fig.1a to clearly state the meaning of XL and the purpose of *E.coli* supernatant. For example, the corresponding main text is revised as follows: "The Kbz signal was readily detected on core histones extracted from logarithmically growing yeast cells, but not on the recombinant *Xenopus laevis* histone octamer purified from *E.coli* and cell lysates from *E.coli* (Fig. 1a)."

2. Figure 1B: Experimental details are missing. For how long the cells were treated with sodium benzoate? Was BY4741 used? This information must be added in figure legends at least.

Thank you for this suggestion. We have extensively revised our method section to include more experimental details and clearly stated some key information in the figure legends.

3. Line 90-92: Especially considering that the antibody used in pan Kbz, the data is not clear enough to make such a conclusion. Speculation is ok. But the basis of the speculation should be described.

Thank you for your suggestion, and we agree that our results do not fully support our original conclusion. Thus we have deleted this sentence in the revision.

4. Figure 1C: Again, very poorly described experiment. What does H₂O mean? Cells were moved from YPD to water? Yeast won't grow in water and will be stagnant. So does that mean that stagnant yeast loses the mark?

We are sorry for not stating this clearly. In the revision, we have included experimental details in the figure legends as follows: "The BY4742 cells were grown in standard medium with 2% glucose to log phase, and then transferred to H₂O or other indicated mediums for 4 hours, followed by extraction of histones and western blotting analysis."

5. Line 102: For how long were the cells treated with sodium benzoate?

We apologize for not including this important information in the initial submission. In the revision, we have included this information in the manuscript as follows: "BY4742 cells were grown in YPD to log phase, and then sodium benzoate was added into the medium for 6 h, followed by extraction of histones and western blotting analysis".

6. Line 105: What happens to H3K9bz, H3K14bz, H3K27bz, and H4K44bz when yeast are grown in 4% EtOH?

We thank the reviewer for raising this issue. We also think it would be great to check the lysine benzoylation signal change at individual histone sites under different conditions. However, due to the lack of site-specific antibodies against lysine benzoylation, we cannot address this issue at this stage. In view of the fact that the Kbz level cultured in 2% EtOH was significantly less than that cultured in 2% glucose (Fig. 1c), we hypothesize that the levels of H3K9bz, H3K14bz, H3K27bz, and H4K44bz will be substantially reduced.

7. Mass spec data: The results need to be more descriptive. How much percentage of peptides were modified? How many replicates?

We thank you for raising this issue. We evaluated the stoichiometry of lysine benzoylation in six replicates on extracted histones from yeast samples without enrichment. The average abundance of lysine benzoylation on histones is around 0.056%, which is much lower than that of other well-known modifications, such as methylation and acetylation. We have included this information in the revised manuscript.

8. Line 167-171: The authors state that 'Removal of Class I HDAC (Rpd3) or Class II HDAC (Hda1, Hos1, Hos2, Hos3) enzymes had no or negligible effect on Kbz signals.' I do not agree with this conclusion as the western blot data in Figure 3a

doesn't support this. Rpd3 Δ almost completely abolishes Kbz. Also, the changes in Kbz signal appear similar in Hos1 and Hos2 deletion as that seen upon hst2 and 3 deletions.

We thank you for this comment. In the revision, we re-performed the western blot assays with the widely used Odyssey Infrared Imaging system, which gives accurate quantitative western blot results with near-infrared fluorescence. Figs. 2a and 3a were updated. In the new Figs. 2a and 3a, the fluorescence signals from mutant strains were normalized by setting the WT fluorescence signal to 1.0 and were labeled in the image. The reviewer is correct. Deletion of Hos1 and Hos2 also mildly increased the benzoylation signals of H3. According to the new quantification data, we have rephrased our description about HDAC-deletion data as follows: "In sharp contrast, deletion of most HDACs only mildly increased the benzoylation level less than 2-fold (Fig. 3a). Removal of Hst2 caused the highest increase in the H3 Kbz signal by approximately 1.8-fold (Fig. 3a). Unexpectedly, the removal of some HDACs, including Rpd3, Hos3, and Sir2, decreased the global histone benzoylation levels (Fig. 3a). These enzymes might be involved in the benzoyl-CoA metabolic pathway or regulate expression levels of histone benzoyltransferases, which merits further investigation."

9. Figure 3b-f: Using peptides in in vitro assays the authors profile the substrate specificity of Hst2-mediated debenzoylation. This data should be completed with mass spec analysis of histones from the Hst2 Δ BY4742 strain as well. In fact, many of the in vitro assays presented in this manuscript to test activity will be benefited by complementing them with experiments done in yeast cells.

We thank the reviewer for this suggestion. Unfortunately, the MS data from our current procedure have great variation from batch to batch due to the low abundance of lysine benzoylation, thus preventing us from reaching a conclusive conclusion by simply comparing MS spectra from WT and *Hst2 Δ* strains. To accurately quantify the benzoylation change by MS analyses, SILAC is required for absolute amount comparison, which needs extensive time to be performed and analyzed. Thus, we choose not to include these data in the current manuscript. We hope this reviewer will understand our choice.

10. Line 186-188: Citing the data shown in figure 1, the authors conclude that H3 is the most extensively benzoylated histone among core histones. However, based on figure 1 this cannot be concluded. What percentage of peptides were modified? The results from western blot might be due to the lower affinity of the antibody for H4.

We agree with this reviewer's comment that, due to the different specificity and affinity of pan-benzoyl lysine toward different proteins, the signal intensity of the western blot is not comparable between different benzoylation sites or between H3 and H4. In the revision, we have deleted this unsupported claim and revised this

sentence as follows: "We synthesized a panel of histone peptides containing different benzoylation sites on histone H3".

11. Line 199-201: The authors state that 'critical arginine residue adjacent to the target lysine site determining the tight binding between Hst2 and histone substrates'. The authors should mutate R and do ITC to provide direct evidence for this. Also, what is the importance of R structurally? Should be discussed.

We thank the reviewer for this suggestion. Structurally, H3 R8 is oriented to the acidic patch composed of Hst2 D68 and D70. Thus, mutation of the residue is predicted to decrease the Hst2-H3 interaction. In the revision, we have included the ITC data for Hst2 with the H3K9bz^{R8A} peptide (Fig. 4e). The data showed that the binding affinity between the H3K9bz^{R8A} peptide and Hst2 was decreased about 3.3-fold. The debenzoylation activity of Hst2 on this mutant H3 peptide was also reduced (Fig. 4f).

12. Line 306-309: The authors state that 'Sas5^{YEATS} may be a favorable Kbz reader in yeast cells'. Again, no in vivo data to support this.

Thank you for this suggestion. Our ITC data show that Taf14^{YEATS} binds Kbz at an affinity similar to that of Kac and Kcr, but Sas5^{YEATS} displayed a nearly 4-fold higher affinity for Kbz than Kac. Therefore, we initially claimed that "Sas5^{YEATS} may be a favorable Kbz reader in yeast cells." We agree with this reviewer that this speculation needs more in vivo data. Thus, we have modified our tone as follows: "Moreover, Sas5^{YEATS} showed a binding preference for Kbz over Kac (Supplementary Fig. 5c), suggesting that Sas5^{YEATS} might serve as a Kbz reader in yeast cells".

13. Line 416-419: Again, the important information of how much percentage of peptides were modified is missing.

Thank you for this suggestion. According to the quantitative analysis by spectral counting of the whole cell sample with enrichment by a pan-Kbz antibody, around 1.5% of the peptides were detected to be benzoylated. However, this number is from the pan anti-Kbz antibody enriched sample and cannot reflect the real ratio of benzoylated peptides in the whole-cell proteome. We did not perform the Kbz-proteomic analyses without enrichment because the abundance of Kbz was too low in unenriched samples. We apologize that we cannot provide this information at this stage.

14. Line 438-440: In the absence of any direct evidence, this statement is just speculation and should not be phrased as a conclusion.

Thanks for this suggestion. In the revision, we have revised this sentence as follows: "Therefore, these results indicate that lysine benzoylation is a widespread modification in yeast cells and implicate a potential link between lysine benzoylation

modification and cell metabolic pathways."

15. Figure 2a: Why there is an increase in Kbz in the Hpa1 deletion strain? Should be discussed.

Thank you for raising this issue. In the revision, we re-performed the quantitative western blot assays. The updated Fig. 2a showed that deletions of most HATs, including Hpa2, caused a mild increase in Kbz signals (1.2-1.7-fold). Considering that deletions of these HATs decreased the acetylation signals as expected, these deletion strains were correctly constructed. We do not have a clear answer for this observation at this stage. One possibility is that these HAT-deletion may compensatorily up-regulate benzoylase Gcn5 expression and increase Kbz levels. The other possibility is that these HAT-deletion may alter the cellular metabolic state and probably increase benzoyl-CoA concentration. We have revised this part as follows: "Except for Gcn5, deletions of other HATs caused unexpected mild increase of Kbz signal (Fig. 2a). Whether this is due to experimental variation and what is the underlying mechanism for such a Kbz increase need further investigation."

16. Figure 2b: Why band shift in anti-H3N samples?

Thank you for raising this issue. The shift of H3N bands may be caused by the voltage instability during the running of SDS-PAGE. In the revision, we have re-run the gel for the original Fig. 2b, and the new data are now presented in Supplementary Fig. 3c.

17. Figure 2b and 2c: There is no control that lacks Ada2 Gcn5 as well as SAGA.

We thank this reviewer's insightful suggestion. According to this reviewer's suggestion, we have compared the benzoylation reaction in the presence and absence of Gcn5-Ada2. The data was shown in new Fig. 2b-c and also offered as Reference Fig. 3 (in the reply for reviewer 2) for your reference. In the absence of enzymes, we could still see increased benzoylation signals, indicating spontaneous deposition of Kbz by benzoyl-CoA. The LC-MS/MS data also confirmed the western blot data that histone benzoylation could spontaneously occur on histones in the absence of enzymes, but Gcn5-Ada2 could catalyze more benzoylated histone sites and at higher abundances (Fig. 2d-e). Therefore, we propose that lysine benzoylation can occur by both non-enzymatic and enzyme-dependent mechanisms.

18. Figure 2b and 2c: Which histone sites are being modified by Ada2 Gcn5 and SAGA? Mass spec data will be useful.

We thank the reviewer's suggestion. According to this suggestion, we have used LC-MS/MS to identify the Kbz sites specifically catalyzed by Gcn5-Ada2. A summary of the lysine benzoylation sites catalyzed by Gcn5-Ada2 is shown in Fig. 2d.

The representative MS/MS spectra of H3K14bz is shown in Supplementary Fig. 3k. The complete list of histone benzoylation sites modified by Gcn5-Ada2 can be found in Supplementary Table 1.

19. Figure 3a: Many of the differences are subtle. The experiment should be repeated and quantification plots should be added. Also, the H4 signal appears to be saturated. A lower exposure blot would be appreciated.

We thank the reviewer's suggestion. In the revision, we re-performed the experiment using the widely-used Odyssey Infrared Imaging system, which gives accurate quantitative western blot results. The quantification data are presented in the new Figs. 2a and 3a.

20. Figure 3a: Why does sir2 deletion result in a decrease in Kbz? Discussion or speculation would be welcome.

We thank the reviewer's comments. We have revised the description of this phenomenon as follows: " Unexpectedly, the removal of some HDACs, including Rpd3, Hos3, and Sir2, decreased the global histone benzoylation levels (Fig. 3a). These enzymes might be involved in the benzoyl-CoA metabolic pathway or regulate the expression levels of histone benzoyltransferases, which merit further investigation ".

21. Figure 3e: Unmodified H3 peptide should be included as control I the ITC experiment.

We thank the reviewer for this suggestion and have provided ITC data for unmodified H3P30 binding to three YEATS domains. It clearly showed that unmodified H3P30 has almost no affinity with three YEATS domains (Fig. 5a-c).

22. Figure 5a and b: Based on the crystal structure, please explain why the binding specificities might be different.

Thank you for this suggestion. In the previous submission, we only focused on the binding pocket for acyl groups. In the revision, we have provided new Supplementary Fig. 5h-i to dissect the interaction networks between Taf14_{YEATS} and H3K9bz and between Sas5_{YEATS} and H3K27bz. We also provide a more detailed structural description to explain the binding specificities of Taf14_{YEATS} and Sas5_{YEATS} as follows: "In the Taf14_{YEATS}-H3K9bz structure, H3 R8 forms salt bridges with Taf14 D104 (Supplementary Fig. 5k) and mutation of R8 severely disrupted the interaction between Taf14_{YEATS} and H3K9bz (Supplementary Fig. 5b), reinforcing the importance of the neighboring arginine in ensuring binding specificity between Taf14_{YEATS} and histone peptides containing "RK" signatures (H3K9, H3K18, and H3K27) (Fig. 5a). In the Sas5_{YEATS}-H3K27bz structure, in addition to the K27bz-mediated interactions,

H3 A29 and P30 make extensive hydrophobic contacts with a hydrophobic groove composed of Sas5 L28, R31, W82, and F108 (Supplementary Fig. 5I). The downstream "AP" signatures are only observed in H3K14 and H3K27 peptides, explaining the relatively higher binding affinities with H3K14 and H3K27 peptides than other peptides (Fig. 5b)".

23. Figure 5C: Again, no evidence was provided from experiments done in yeast.

We agree with this comment. In the present study, we only provide in vitro evidence that purified Yaf9_{YEATS} have no detectable Kbz-binding ability. Whether Yaf9 could or could not serve as a Kbz reader needs more in vivo data. Thus, we have modified our tone as follows: "In sharp contrast, Yaf9_{YEATS} exhibited no detectable binding to any tested Kbz peptide in our assay condition (Fig. 5c), although Yaf9_{YEATS} could bind to H3K27ac peptide with the similar K_d (K_d=280μM) as reported previously²⁶(Supplementary Fig. 5a). These results demonstrate that the yeast YEATS domains from Taf14 and Sas5 are potential readers for benzoylated lysines and show different substrate preferences in vitro."

24. Does treatment of yeast cells with sodium benzoate alter global H3 acetylation level?

We thank the reviewer's suggestion. In the revision, we have shown that sodium benzoate treatment also significantly enhanced the Kac signals on yeast histones in a dose-dependent manner, similar to the Kbz signals (Fig. 1b). This is in sharp contrast to observations in mammalian cells. In human HepG2 and RAW cells, sodium benzoate treatment had little effect on Kac levels⁵. We speculate that sodium benzoate is an anti-fungal reagent and causes dramatic cellular stress and metabolic changes in yeast cells, consequently inducing the upregulation of Kac signals. The other possible explanation is that yeast and mammalian cells may have different benzoyl-CoA metabolic pathways. In yeast cells, the benzoate-stimulated benzoyl-CoA might be easily degraded by an initial reduction of the aromatic ring, followed by ring hydrolysis, to ultimately form small molecules, such as acetyl-CoA⁶. Thus, the increase of Kac might be an additional consequence of the benzoyl-CoA enhancement. The exact mechanism waits for further investigation.

25. The structural work by the authors reveals that the mode of binding to acetyl and benzoyl groups is very similar. Do the acetyl and benzoyl groups compete for binding?

This is absolutely correct. Actually, all the "reader" proteins use the same binding pockets to recognize acyl groups (acetyl, benzoyl, crotonyl, succinyl, etc.), and their bindings are incompatible.

Minor comments:acyl-

1. The manuscript will be benefitted from thorough proofreading.

We thank the reviewer's suggestion. The revised manuscript has been professionally edited by a native speaker.

a. Line 41: Was initially identified, not is.

Thanks. We have corrected it.

b. Line 56: is unclear, not are.

Thanks. We have corrected it.

c. Line 62: is, not are.

Thanks. We have corrected it.

d. Line 64: in, not from.

Thanks. We have corrected it.

e. Line 77: performed, not perform.

Thanks. We have corrected it.

f. Line 944: series, not serial

Thanks. We have corrected it.

g. Line 257: examined, not examine

Thanks. We have corrected it.

h. Line 267: ensures, not assures

Thanks. We have corrected it.

2. Mention Kac stands for lysine acetylation.

Thanks. We have double-checked the manuscript to spell out the full term when the abbreviation first appears.

3. Line 52: I think at this stage of the manuscript, it is better to replace histone acetyltransferase with enzymes. Because otherwise, it shows an inherent bias that HATs will be responsible for histone benzoylation.

Thanks. This sentence has been revised as follows: "Histone acylation can be non-enzymatically deposited by chemically reactive metabolites or catalyzed by enzymes (e.g., histone acetyltransferases).".

4. Line 68-71: Why suddenly talk about Khib? I feel that it is distracting.

Sorry for distracting in reading this part. According to your suggestion, we have revised this part as follows: " Other forms of lysine acylation, including succinylation (Ksucc), 2-hydroxyisobutyrylation (Khib), β -hydroxybutyrylation (Kbhb), and malonylation (Kmal), are also commonly identified on non-histone proteins^{10, 11, 16, 17}."

5. There should be a separate section for antibodies under Material and Methods.

Thanks. In the revision, we have included a new section in Methods to list all the antibodies used in this manuscript.

6. Figures overall, especially the font size should be bigger for easy readability.

Thanks for the suggestion. In the revision, we have updated labels in all the figures to ensure that readers can clearly see them at the printed size.

7. Line 162: Use of the phrase 'deletion mutants' is more common in yeast than 'knock-out'.

Thanks. We have corrected it.

8. Line 420-424: How was this determined? Which software or server was used?

Thanks. Subcellular distribution analysis was performed by Uniport (<https://www.uniprot.org/>). In the revision, we have included the information in the Methods section.

9. Figure 4f should come after 4C. However, this is an optional change.

Thanks. We have updated Fig. 4 according to the reviewer's suggestion.

10. Line 173: Purified from where?

Thanks. *Saccharomyces cerevisiae* Hst2 proteins were purified from *Escherichia coli* Rosetta cells. We have mentioned the information in the Methods section.

11. Line 289-290: This sentence should be phrased better.

Thanks. We have revised this sentence as follows: "we purified recombinant YEATS domains of Taf14, Sas5, and Yaf9 from *E. coli*."

- 1 Cai, L., Sutter, B. M., Li, B. & Tu, B. P. Acetyl-CoA induces cell growth and proliferation by promoting the acetylation of histones at growth genes. *Mol Cell* **42**, 426-437, doi:10.1016/j.molcel.2011.05.004 (2011).
- 2 Li, X. *et al.* Overproduction of fatty acids in engineered *Saccharomyces cerevisiae*. *Biotechnol Bioeng* **111**, 1841-1852, doi:10.1002/bit.25239 (2014).
- 3 Han, Z. *et al.* Revealing the protein propionylation activity of the histone acetyltransferase MOF (males absent on the first). *J Biol Chem* **293**, 3410-3420, doi:10.1074/jbc.RA117.000529 (2018).
- 4 Sabari, B. R. *et al.* Intracellular Crotonyl-CoA Stimulates Transcription through p300-Catalyzed Histone Crotonylation. *Molecular Cell* **58**, 203-215, doi:10.1016/j.molcel.2015.02.029 (2015).
- 5 Huang, H. *et al.* Lysine benzoylation is a histone mark regulated by SIRT2. *Nat Commun* **9**, 3374, doi:10.1038/s41467-018-05567-w (2018).
- 6 Porter, A. W. & Young, L. Y. Benzoyl-CoA, a universal biomarker for anaerobic degradation of aromatic compounds. *Adv Appl Microbiol* **88**, 167-203, doi:10.1016/B978-0-12-800260-5.00005-X (2014).

REVIEWERS' COMMENTS

Reviewer #1 (Remarks to the Author):

Wang et al. have done a very nice job overall addressing the concerns in the revised manuscript. There is still a lingering concern about the biological relevance of the Kbz modification, especially in light of the new data provided by the authors that only about 0.06% of histones are benzylated and that the rate of benzylation in cells is only 1/520 of that of acetylation. Nonetheless, this comprehensive and rigorous study is an important contribution to the field and sets the stage for further biological and functional exploration.

Minor criticisms that should be addressed before publication are listed below.

1. Line 58, suggest changing “vague” to “unclear”
2. Line 62, add “domains to the end of the sentence.
3. Line 66, change “tremendous” to a tremendous number of”
4. Line83, change “foundations” to “the foundation”
5. Line 113, suggest deleting the word “dynamic”
6. Line 211-214, the following supporting publication should be mentioned and referenced:
<https://www-nature-com.proxy.library.upenn.edu/articles/s41467-018-05567-w>
7. Line 310, change “V182” to “V182A”
8. Line 325, suggest changing “prove” to “support”

Reviewer #2 (Remarks to the Author):

The study by Wang et.al. characterizes lysine benzylation in *S. cerevisiae*, a recently discovered post translational modification in mammalian cells. The authors use proteomic methods to identify novel lysine benzylation sites on histone and non-histone proteins. Additionally, biochemical and genetic methods were used to identify the "readers", "writers" and "erasers" of lysine benzylation. The methods used in this study were sound and the experimental controls were appropriate for this study. This reviewer addressed a few concerns in the first round of revisions and I think the authors addressed those concerns. The resulting work is of high quality and will be impactful for the field.

Reviewer #3 (Remarks to the Author):

The authors have done a decent job in addressing my concerns overall and incorporating the changes that I suggested. The manuscript still continues to lack enough *in vivo* data. However, considering the novelty of the findings and the strong *in vitro* data, I feel that the manuscript should be published in Nature Communications.

REVIEWERS' COMMENTS

Reviewer #1 (Remarks to the Author):

Wang et al. have done a very nice job overall addressing the concerns in the revised manuscript. There is still a lingering concern about the biological relevance of the Kbz modification, especially in light of the new data provided by the authors that only about 0.06% of histones are benzylated and that the rate of benzylation in cells is only 1/520 of that of acetylation. Nonetheless, this comprehensive and rigorous study is an important contribution to the field and sets the stage for further biological and functional exploration.

Minor criticisms that should be addressed before publication are listed below.

We thank this reviewer for the appreciation of our work and constructive suggestions.

1. Line 58, suggest changing “vague” to “unclear”

Thanks. We have corrected it.

2. Line 62, add “domains to the end of the sentence.

Thanks. We have corrected it.

3. Line 66, change “tremendous” to a tremendous number of”

Thanks. We have corrected it.

4. Line83, change “foundations” to “the foundation”

Thanks. We have corrected it.

5. Line 113, suggest deleting the word “dynamic”

Thanks. We have corrected it.

6. Line 211-214, the following supporting publication should be mentioned and referenced: <https://www-nature-com.proxy.library.upenn.edu/articles/s41467-018-05567-w>

Thanks. We have referenced this paper.

7. Line 310, change “V182” to “V182A”

Thanks. We have corrected it.

8. Line 325, suggest changing “prove” to “support”

Thanks. We have corrected it.